# Age-dependent topic modeling of comorbidities in UK Biobank identifies disease subtypes with differential genetic risk

Xilin Jiang [1,2,3,4,5,6] ✉, Martin Jinye Zhang [4,7,18], Yidong Zhang[1,8,9,18], Arun Durvasula[4,7,10,11,18], Michael Inouye[5,6,12,13,14,15,16], Chris Holmes[1,2,16], Alkes L. Price [4,7,17,19] ✉ & Gil McVean [1,19] ✉

The analysis of longitudinal data from electronic health records (EHRs) has the potential to improve clinical diagnoses and enable personalized medicine, motivating efforts to identify disease subtypes from patient comorbidity information. Here we introduce an age-dependent topic modeling (ATM) method that provides a low-rank representation of longitudinal records of hundreds of distinct diseases in large EHR datasets. We applied ATM to 282,957 UK Biobank samples, identifying 52 diseases with heterogeneous comorbidity profiles; analyses of 211,908 All of Us samples produced concordant results. We defined subtypes of the 52 heterogeneous diseases based on their comorbidity profiles and compared genetic risk across disease subtypes using polygenic risk scores (PRSs), identifying 18 disease subtypes whose PRS differed significantly from other subtypes of the same disease. We further identified specific genetic variants with subtype-dependent effects on disease risk. In conclusion, ATM identifies disease subtypes with differential genome-wide and locus-specific genetic risk profiles.

Longitudinal electronic health record (EHR) data, encompassing diagnoses across hundreds of distinct diseases, offers immense potential to improve clinical diagnoses and enable personalized medicine[1]. Despite intense interest in both the genetic relationships between distinct diseases[2–11] and the genetic relationships between biological subtypes of disease[12–15], there has been limited progress on classifying disease phenotypes into groups of diseases with frequent co-occurrences (comorbidities) and leveraging comorbidities to identify disease subtypes. Low-rank modeling has appealing theoretical properties[16,17] and has produced promising applications[18–24] to infer meaningful representations of high-dimensional data. In particular, low-rank representation is an appealing way to summarize data across hundreds of distinct diseases[25–27], providing the potential to identify patient-level

comorbidity patterns and distinguish disease subtypes. The biological differentiation of disease subtypes inferred from EHR data could be validated by comparing genetic profiles across subtypes, which is possible with emerging datasets that link genetic data with EHR data[28–31].

Previous studies have used low-rank representation to identify shared genetic components[25–27] across multiple distinct diseases, identifying relationships between diseases and generating valuable biological insights. However, age at diagnosis information in longitudinal EHR data has the potential to improve such efforts. For example, a recent study used longitudinal disease trajectories to identify disease pairs with statistically significant directionality[32], suggesting that age information could be leveraged to infer comorbidity profiles that capture temporal information. In addition, patient-level comorbidity

**Fig. 1 | ATM provides an efficient way to represent longitudinal comorbidity data.** Top left, input consists of disease diagnoses as a function of age. Top right, ATM assigns a topic weight to each patient. Bottom, ATM infers age-dependent topic loadings.

information could potentially be leveraged to identify biological subtypes of the disease, complementing its application to increase power for identifying genetic associations[12] and to cluster disease-associated variants into biological pathways[8]; disease subtypes are fundamental to disease etiology[14,33–36].

Here we propose an age-dependent topic modeling (ATM) method to provide a low-rank representation of longitudinal disease records. ATM learns, and assigns to each individual, topic weights for several disease topics, each of which reflects a set of diseases that tend to co-occur within individuals as a function of age. We applied ATM to 1.7 million disease diagnoses spanning 348 diseases in the UK Biobank, inferring ten disease topics; we validated ATM in All of Us. We identified 52 diseases with heterogeneous comorbidity profiles that enabled us to define disease subtypes. We used genetic data to validate the disease subtypes, showing that they exhibit differential genome-wide and locus-specific genetic risk profiles.

## Results

### Overview of methods

We propose an ATM model, which provides a low-rank representation of longitudinal records of hundreds of distinct diseases in large EHR datasets (Fig. 1; Methods). The model assigns to each individual

'topic weights' for several 'disease topics'. Each disease topic reflects a set of diseases that tend to co-occur as a function of age, quantified by age-dependent 'topic loadings' for each disease. The model assumes that, for each disease diagnosis, a topic is sampled based on the individual's topic weights (which sum to 1 across topics, for a given individual), and a disease is sampled based on the individual's age and the age-dependent topic loadings (which sum to 1 across diseases, for a given topic at a given age). The model generalizes the latent Dirichlet allocation (LDA) model[37,38] by allowing topic loadings for each topic to vary with age (Supplementary Note and Extended Data Fig. 1).

We developed a method to fit this model that addresses several challenges inherent to large EHR datasets. The method estimates topic weights for each individual, topic loadings for each disease and posterior diagnosis-specific topic probabilities for each disease diagnosis. First, we derived a scalable deterministic method that uses numerical approximation approaches to fit the parameters of the model, addressing the challenge of computational cost. Second, we used the prediction odds ratio[39] to compare model structures (for example, number of topics and parametric form of topic loadings as a function of age), addressing the challenge of appropriate model selection; roughly, the prediction odds ratio quantifies the accuracy of correctly predicting disease diagnoses in held-out individuals using comorbidity

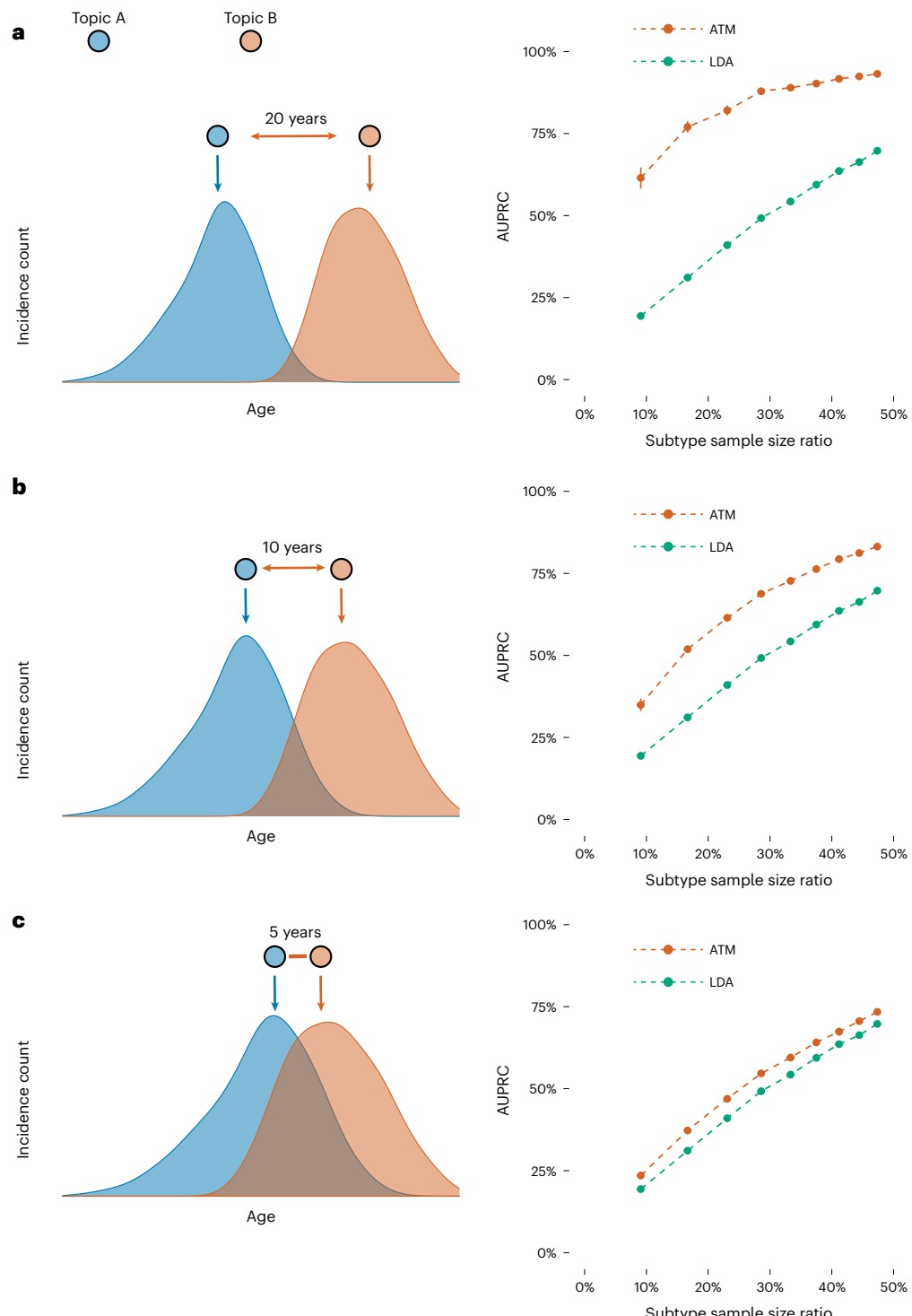

**Fig. 2 | ATM outperforms LDA in simulations with age-dependent effects.**
In simulations at different levels of age-dependent effects (left), we report the
AUPRC for ATM versus LDA as a function of subtype sample size proportion
(the proportion of diagnoses belonging to the smaller subtype; right). Each dot
represents the mean of 100 simulations of 10,000 individuals. Error bars denote
95% confidence intervals. **a**, Twenty-year difference in age at diagnosis for the
two subtypes. **b**, Ten-year difference in age at diagnosis for the two subtypes.
**c**, Five-year difference in age at diagnosis for the two subtypes. Numerical results
are reported in Supplementary Table 2.

information, compared to a predictor based only on prevalence (Methods; Supplementary Table 1). Third, we used collapsed variational inference[40], addressing the challenge of sparsity in the data (for example, in UK Biobank data, the average patient has diagnoses for 6 of 348 diseases analyzed); collapsed variational inference outperformed mean-field variational inference[37] in empirical data. Further details are provided

in the Methods and Supplementary Note; we have publicly released open-source software implementing the method (Code availability).

We applied ATM to longitudinal records of UK Biobank[29] (282,957 individuals with 1,726,144 disease diagnoses spanning 348 diseases; the targeted individuals are those diagnosed with at least two of the 348 diseases studied) and All of Us[30] (211,908 individuals with 3,098,771

**Table 1 | Summary of ten inferred disease topics in the UK Biobank**

| Acronyms | Disease systems | Representative diseases | Number of associated diseases |
|---|---|---|---|
| NRI | Neoplasms, respiratory, infectious diseases | Secondary malignancy of lymph nodes; pneumococcal pneumonia; bacterial infection NOS | 53 |
| CER | Circulatory system, endocrine/metabolic, respiratory | Type 2 diabetes; obesity; chronic airway obstruction | 41 |
| SRD | Sense organs, respiratory, dermatologic | Cataract; septal deviations/turbinate hypertrophy; benign neoplasm of skin | 38 |
| CVD | Cardiovascular disease | Hypercholesterolemia; coronary atherosclerosis; myocardial infarction | 27 |
| UGI | Upper gastrointestinal disease | Diaphragmatic hernia; benign neoplasm of other parts of digestive system; gastritis and duodenitis | 22 |
| LGI | Lower gastrointestinal disease | Irritable bowel syndrome; benign neoplasm of colon; anal and rectal polyp | 13 |
| FGND | Female genitourinary, neoplasms, digestive | Uterine leiomyoma; malignant neoplasm of female breast; hypothyroidism NOS | 34 |
| MGND | Male genitourinary, neoplasms, digestive | Urinary tract infection; cancer of prostate; other disorders of bladder | 33 |
| MDS | Musculoskeletal, digestive, symptoms | Back pain; cholelithiasis; other disorders of soft tissues | 29 |
| ARP | Arthropathy-related disease | Arthropathy NOS; rheumatoid arthritis; enthesopathy | 26 |

For each topic, we list its three-letter acronym, disease systems, representative diseases and number of associated diseases (defined as diseases with average diagnosis-specific topic probability >50% for that topic). Topics are ordered by the Phecode system (Fig. 3). In total, 316 of 348 diseases analyzed are associated with a topic; the remaining 32 diseases do not have a topic with average diagnosis-specific topic probability >50%. NOS, not otherwise specified.

disease diagnoses spanning 233 of the 348 diseases). Each disease diagnosis has an associated age-at-diagnosis, defined as the earliest age of reported diagnosis of the disease in that individual; we caution that age at diagnosis may differ from age at disease onset (Discussion). ATM does not use genetic data, but we used genetic data to validate the inferred topics (Methods).

## Simulations

We performed simulations to compare ATM with LDA[37,38], a simpler topic modeling approach that does not model age. Choices of simulation parameters that resemble real data are described in Supplementary Note. We assigned each disease diagnosis to one of two subtypes for the target disease based on age and other subtype differences, considering high, medium or low age-dependent effects by specifying an average difference of 20, 10 or 5 years, respectively, in age at diagnosis for the two subtypes. For each level of age-dependent effects, we varied the proportion of diagnoses belonging to the first subtype (that is, the subtype that has an earlier average age-at-diagnosis) from 10% to 50%. Our primary metric for evaluating the LDA and ATM methods is the area under the precision–recall curve (AUPRC)[41] metric, where precision is defined as the proportion of disease diagnoses that a given method assigned to the first subtype that was assigned correctly, and recall is defined as the proportion of disease diagnoses truly belonging to the first subtype that was assigned correctly. We discretized the subtype assigned to each disease diagnosis by a given method by assigning the subtype with higher inferred probability. We used AUPRC (instead of prediction odds ratio) in our simulations because the underlying truth is known. Further details and justifications of metrics used in this study are provided in the Methods, Supplementary Note and Supplementary Table 1.

In simulations with high age-dependent effects, ATM attained much higher AUPRC than LDA across all values of subtype sample size proportion (AUPRC difference: 24–42%), with both methods performing better at more balanced ratios (Fig. 2 and Supplementary Table 2). Accordingly, ATM attained both higher precision and higher recall than LDA (Supplementary Fig. 1). Results were qualitatively similar when using the second subtype as the classification target (Supplementary Fig. 2). In simulations with medium or low age-dependent

effects, ATM continued to outperform LDA, but with smaller differences between the methods. In simulations without age-dependent effects, ATM slightly underperformed LDA (Supplementary Fig. 3a). Three secondary analyses are described in the Supplementary Note and Supplementary Figs. 3 and 4.

We conclude that ATM (which models age) assigns disease diagnoses to subtypes with higher accuracy than LDA (which does not model age) in simulations with age-dependent effects. We caution that our simulations largely represent a best-case scenario for ATM given that the generative model and inference model are very similar (although there are some differences, for example, topic loadings were generated using a model different from the inference model), thus it is important to analyze empirical data to validate the method.

## Age-dependent comorbidity profiles in the UK Biobank

We applied ATM to longitudinal hospital records of 282,957 individuals from the UK Biobank with an average record span of 28.6 years[29]. We used Phecode[42] to define 1,726,144 disease diagnoses spanning 348 diseases with at least 1,000 diagnoses each. The average individual had 6.1 disease diagnoses, and the average disease had a s.d. of 8.5 years in age at diagnosis. The optimal inferred ATM model structure has ten topics and models age-dependent topic loadings for each disease as a spline function with one knot, based on optimizing prediction odds ratio (see below). We assigned names (and corresponding acronyms) to each of the ten inferred topics based on the Phecode systems[42] assigned to diseases with high topic loadings (aggregated across ages) for that topic (Table 1 and Supplementary Table 3).

Age-dependent topic loadings across all ten topics and 348 diseases (stratified into Phecode systems), summarized as averages across age <60 years and age ≥60 years, are reported in Fig. 3, Extended Data Fig. 2 and Supplementary Table 4. Some topics such as neoplasms, respiratory, infectious diseases (NRI) span diseases across the majority of Phecode systems, while other topics such as arthropathy-related disease (ARP) are concentrated in a single Phecode system. Conversely, a single Phecode system may be split across multiple topics, for example, diseases of the digestive system are split across upper gastrointestinal disease (UGI), lower gastrointestinal disease (LGI) and musculoskeletal, digestive, symptoms (MDS). We note that topic loadings in diseases that

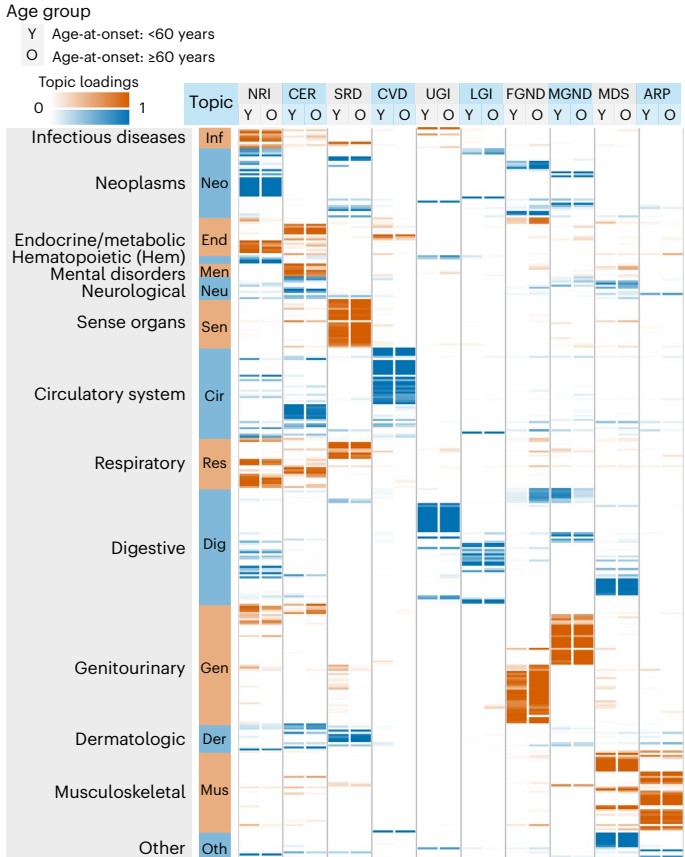

**Age group**
Y Age-at-onset: <60 years
O Age-at-onset: ≥60 years

Topic loadings
0 — 1

**Fig. 3 | Age-dependent topic loadings of ten inferred disease topics across 348 diseases in the UK Biobank.** We report topic loadings averaged across younger ages (age at diagnosis <60 years) and older ages (age at diagnosis >60 years). Row labels denote disease categories ordered by Phecode systems, with alternating blue and red colors for visualization purposes; 'Other' is a merge of five Phecode systems: 'congenital anomalies', 'symptoms', 'injuries and poisoning', 'other tests' and 'death' (which is treated as an additional disease; Methods). Topics are ordered by the corresponding Phecode system. Further details on the ten topics are provided in Table 1. Further details on the diseases discussed in the text (type 2 diabetes and breast cancer) are provided in Extended Data Fig. 2. Numerical results are reported in Supplementary Table 4.

span multiple topics are heavily age-dependent. For example, patients with type 2 diabetes assigned to the cardiovascular disease (CVD) topic are associated with early onset of type 2 diabetes, whereas patients with type 2 diabetes assigned to the male genitourinary, neoplasms, digestive (MGND) topic are associated with late onset of type 2 diabetes.

We performed seven secondary analyses to validate the integrity and reproducibility of inferred comorbidity topics. First, we fit ATM models with different model structures using 80% training data and computed their prediction odds ratios using 20% testing data. The ATM model structure with ten topics and age-dependent topic loadings modeled as a spline function performed optimally (Supplementary Fig. 5; Methods). Second, we confirmed that ATM (which models age) attained higher prediction odds ratios than LDA (which does not model age) across different values of the number of topics (Extended Data Fig. 3). For the optimal model with ten topics, ATM attained an average prediction odds ratio of 1.71, compared to a prediction odds ratio of 1.58 for LDA. Third, we compared the topic loadings by repeating the inference on female-only or male-only populations and observed no major discrepancies, except for genitourinary topics MGND and female genitourinary, neoplasms, digestive (FGND; topic loading $R^2$ (female versus all) = 0.788, topic loading $R^2$ (male versus all) = 0.773; Extended Data Fig. 4). Fourth, we verified that body mass index (BMI),

sex, Townsend deprivation index and birth year explained very little of the information in the inferred topics (Supplementary Table 3). Three additional secondary analyses are described in the Supplementary Note, Supplementary Figs. 6–8 and Supplementary Table 1.

Disease topics capture known biology and the age-dependency of comorbidities for the same diseases. For example, the early onset of essential hypertension is associated with the CVD topic[43], which captures the established connection between lipid dysfunction (hypercholesterolemia) and CVDs[44], whereas the later onset of essential hypertension is associated with the circulatory system, endocrine/metabolic, respiratory (CER) topic, which pertains to type 2 diabetes, obesity and chronic obstructive pulmonary disease (COPD) (Fig. 4a). Continuously varying age-dependent topic loadings for all ten topics, restricted to diseases with high topic loadings, are reported in Supplementary Fig. 9 and Supplementary Table 5. We note that most diseases have their topic loadings concentrated into a single topic (Fig. 4b, Supplementary Fig. 10a and Supplementary Table 4) and that most individuals have their topic weights concentrated into 1–2 topics (Fig. 4c and Supplementary Fig. 10b). For diseases spanning multiple topics (Extended Data Fig. 2 and Supplementary Table 4), the assignment of patients with type 2 diabetes to the CVD topic is consistent with known pathophysiology and epidemiology[45,46] and has been shown in other comorbidity clustering studies, for example, with the β cell and lipodystrophy subtypes described in ref. 35 and the severe insulin-deficient diabetes subtype described in ref. 14, which are characterized by early onset of type 2 diabetes and have multiple morbidities including hypercholesterolemia, hyperlipidemia, and cardiovascular diseases[47]. In addition, early-onset breast cancer and late-onset breast cancer are associated with different topics, for example, NRI and FGND, consistent with known treatment effects for patients with breast cancer that increase susceptibility to infections, especially bacterial pneumonias[48] and hypothyroidism[49]. We conclude that ATM identifies latent disease topics that robustly compress age-dependent comorbidity profiles and capture disease comorbidities both within and across Phecode systems.

## Age-dependent comorbidity profiles in All of Us

To assess the transferability of inferred topics between cohorts, we applied ATM to longitudinal data from 211,908 All of Us samples[30]. We analyzed 3,098,771 diagnoses spanning 233 of the 348 diseases analyzed in UK Biobank for which data were available. The average individual had 14.6 disease diagnoses, and an average disease had a standard deviation of 14.0 years in age at diagnosis. The optimal model for All of Us included 13 topics (Supplementary Figs. 11 and 12a,b and Supplementary Table 6). Most diseases have their topic loadings concentrated into a single topic, and most individuals have their topic weights concentrated into 1–4 topics (Supplementary Fig. 13).

We assessed the concordance between each UK Biobank topic and each All of Us topic by computing the correlation between the respective topic loadings across the 233 diseases analyzed in both datasets (Fig. 5a,b, Supplementary Fig. 14 and Supplementary Table 7). The median correlation between the ten UK Biobank topics and the most similar All of Us topic was 0.54, confirming the qualitative alignment of topic loadings between All of Us and UK Biobank. For example, the topic loadings of CVD and CER topics were qualitatively similar to the most similar All of Us topics (Fig. 5a versus Figure 4a), even though disease prevalences differ between the two cohorts (Supplementary Table 8). When using the optimal All of Us model (13 topics) to predict diagnoses in UK Biobank, we obtained a prediction odds ratio that was significantly larger than 1 (mean = 1.32; jackknife s.e. = 0.0027; Supplementary Fig. 12c). Key differences between All of Us and UK Biobank data are described in the Supplementary Note.

For each of the 233 diseases, we assessed the concordance between UK Biobank and All of Us topic assignments for that disease by computing the correlation between UK Biobank topic assignments and

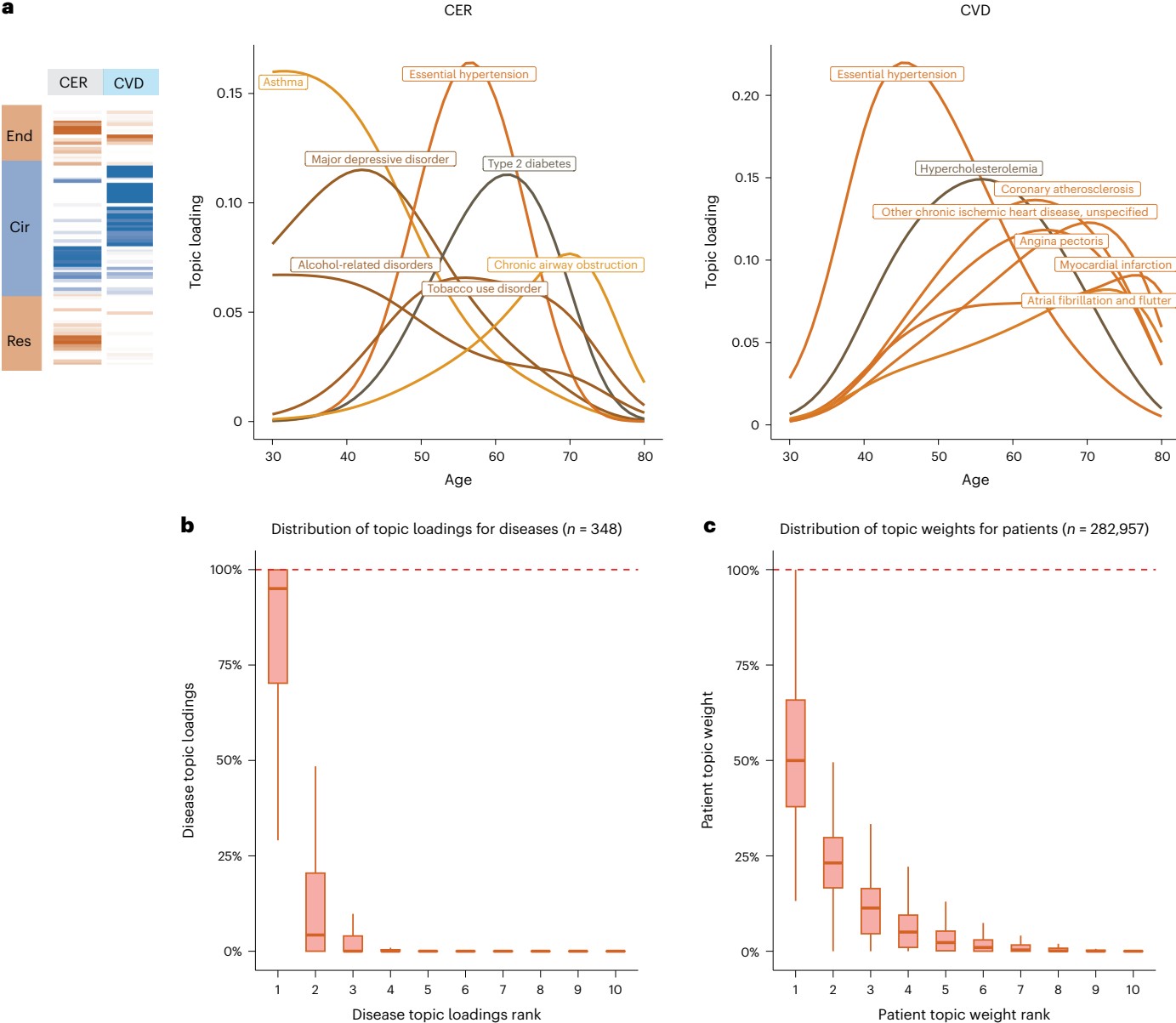

**Fig. 4 | Topic loadings in UK Biobank capture age-dependent comorbidities.** **a**, Age-dependent topic loadings for two representative topics, CER and CVD; for each topic, we include the top seven diseases with the highest topic loadings. Results for all ten topics are reported in Supplementary Fig. 9. **b**, Box plot of disease topic loading as a function of rank; disease topic loadings are computed as a weighted average across all values of age at diagnosis. **c**, Box plot of patient topic weight as a function of rank. Center, box bounds and whisker ends denote median, quartiles and minima/maxima. Numerical results are reported in Supplementary Table 5.

All of Us topic assignments that were mapped to UK Biobank topics (by weighting by correlations between topics; Methods). The average correlation between UK Biobank and All of Us topic assignments for the same disease was 0.70 (versus average correlation of 0.02 for different diseases; Fig. 5c, Extended Data Fig. 5 and Supplementary Table 9). We conclude that ATM identifies latent disease topics from the All of Us cohort that align with topics from the UK Biobank.

**Comorbidity-based subtypes are genetically heterogeneous**

We sought to define disease subtypes in UK Biobank data based on the topic weights of each patient and diagnosis-specific topic probabilities of each disease diagnosis. In some analyses, we used 'continuous-valued topic weights' to model disease subtypes. In analyses that require discrete subtypes, we assigned a discrete topic assignment to each disease diagnosis based on its maximum diagnosis-specific topic probability

and inferred the 'comorbidity-derived subtype' of each disease diagnosis based on the discrete topic assignment; we note that discretizing continuous data loses information (Discussion). We restricted our disease subtype analyses to 52 diseases with at least 500 diagnoses assigned to each of two discrete subtypes; the average correlation between UK Biobank and All of Us disease subtypes (see above; same metric as Fig. 5c) was 0.64 for the 41 (of 52) diseases that were shared between the two cohorts (Methods; Extended Data Fig. 2, Supplementary Fig. 12d and Supplementary Tables 10 and 11).

Age-dependent distributions of comorbidity-derived subtypes for four diseases (type 2 diabetes, asthma, hypercholesterolemia and essential hypertension) are reported in Fig. 6a and Supplementary Table 12; results for all 52 diseases are reported in Supplementary Fig. 16 and Supplementary Table 12, and age-dependent distributions for the same four diseases in All of Us are reported in Supplementary

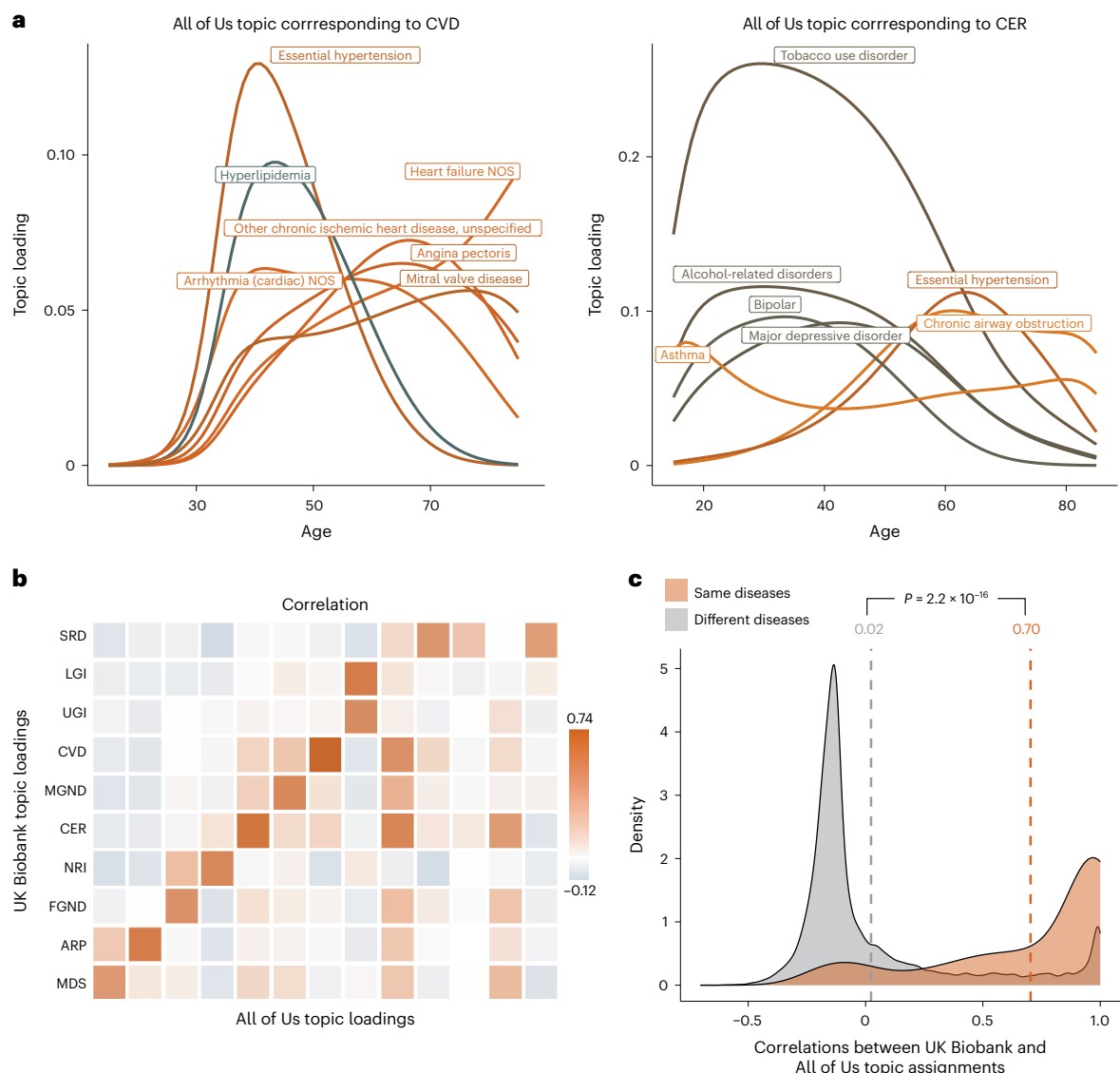

**Fig. 5 | Topic loadings in All of Us capture age-dependent comorbidities that are concordant with UK Biobank. a**, Age-dependent topic loadings for two All of Us topics corresponding to CVD and CER (Fig. 4a); for each topic, we include the top seven diseases with the highest topic loadings. Correlations of topic loadings between UK Biobank and All of Us topics were 0.74 for CVD and 0.65 for CER. Numerical results for all 13 topics are reported in Supplementary Table 6. **b**, Topic loading correlations between UK Biobank and All of Us. The *y* axis reflects the ten topics from the optimal UK Biobank model; the *x* axis reflects the 13 topics from the optimal All of Us model. Numerical results are reported in Supplementary Table 7. **c**, Correlations between UK Biobank and All of Us topic assignments were higher for the same diseases (red shading, average = 0.70) than for different diseases (gray shading, average = 0.02). *P* value is for two-sided *t* test; numerical results are reported in Supplementary Table 9.

Fig. 17. The number of subtypes can be large, for example, six subtypes for essential hypertension. Subtypes are often age-dependent, for example, for the CVD and MGND subtypes of type 2 diabetes[14,35] (discussed above).

ATM and the resulting subtype assignments do not make use of genetic data. However, we used genetic data to assess genetic heterogeneity across inferred subtypes of each disease. We used continuous-valued topic weights in this analysis. We first assessed whether polygenic risk score (PRS) for overall disease risk varied with continuous-valued topic weights for each disease; PRS were computed using BOLT-LMM with fivefold cross-validation[50,51] (Methods and Code availability). Results for four diseases (from Fig. 6a) are reported in Fig. 6b and Supplementary Table 13; results for all ten well-powered diseases (10 of 52 diseases with highest *z*-scores for nonzero SNP-heritability) are reported in Extended Data Fig. 6 and Supplementary Table 13. We identified 18 disease–topic pairs

(of 10 × 10 = 100 disease–topic pairs analyzed) for which PRS values in disease cases vary with patient topic weight. For example, for essential hypertension, hypercholesterolemia and type 2 diabetes, patients assigned to the CVD subtype had significantly higher PRS values than patients assigned to other subtypes. For essential hypertension, patients assigned to the CER subtype had significantly higher PRS values; for type 2 diabetes, patients assigned to the CER subtype had lower PRS values than the CVD subtype, even though the majority of type 2 diabetes diagnoses are assigned to the CER subtype. We further verified that most of the variation in PRS values with disease subtype could not be explained by age[52] or differences in subtype sample size (Supplementary Fig. 18). These associations between subtypes (defined using comorbidity data) and PRS (defined using genetic data) imply that disease subtypes identified through comorbidity are genetically heterogeneous, consistent with phenomenological differences in disease etiology.

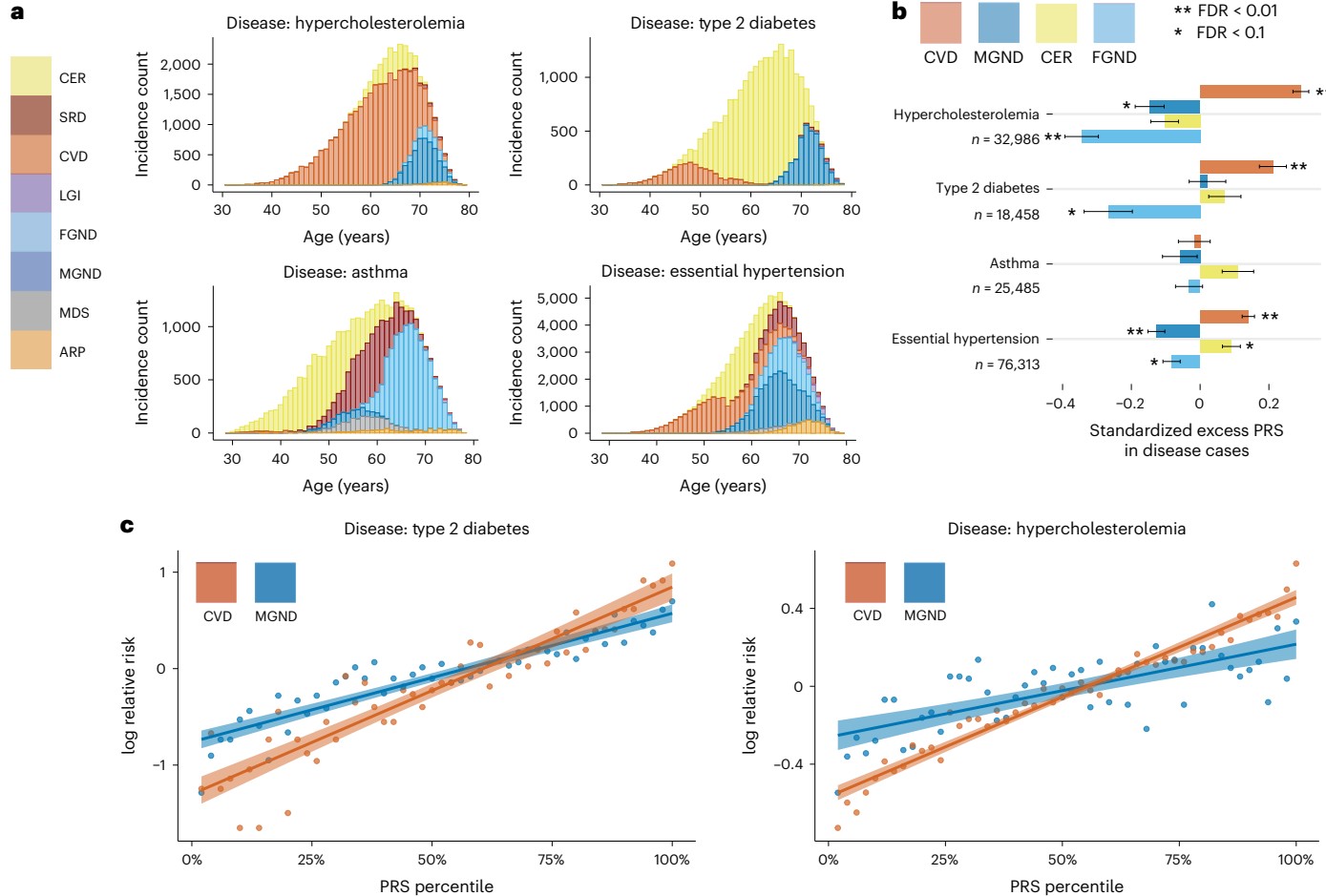

**Fig. 6 | Polygenic risk scores vary across disease subtypes defined by distinct topics. a**, Stacked bar plots of age-dependent subtypes (defined by topics) for four representative diseases (type 2 diabetes, asthma, hypercholesterolemia and essential hypertension); for each disease, we include all subtypes with at least one diagnosis. Results for all 52 diseases are reported in Supplementary Fig. 16. **b**, Standardized excess PRS values in disease cases (s.d. increase in PRS per unit increase in patient topic weight) for four representative diseases and four corresponding topics. **c**, Relative risk for cases of type 2 diabetes and hypercholesterolemia of CVD and MGND subtypes (versus controls) across PRS percentiles. Each point spans two PRS percentiles. Lines denote regression on a log scale. Error bars denote 95% confidence intervals. Numerical results are reported in Supplementary Tables 11–13.

We further investigated whether subtype assignments (defined using comorbidity data) revealed subtype-specific excess genetic correlations. We used discrete subtypes in this analysis. We estimated excess genetic correlations between disease–subtype and subtype–subtype pairs (relative to genetic correlations between the underlying diseases). Excess pairwise genetic correlations for 15 diseases and disease subtypes (spanning 11 diseases and three topics: CER, MGND and CVD) are reported in Fig. 7a and Supplementary Table 14 (relative to genetic correlations between the underlying diseases; Fig. 7b), and excess pairwise genetic correlations for all 89 well-powered diseases and disease subtypes (89 of 378 diseases and disease subtypes with $z$-score >4 for nonzero SNP-heritability; 378 = 348 diseases + 30 disease subtypes) are reported in Supplementary Fig. 19 and Supplementary Table 14. Genetic correlations between pairs of subtypes involving the same disease were significantly less than 1 (false discovery rate (FDR) < 0.1) for hypertension (CER versus CVD: $\rho = 0.86 \pm 0.04$, $P = 0.0004$; MGND versus CVD: $\rho = 0.74 \pm 0.05$, $P = 3 \times 10^{-8}$) and type 2 diabetes (CER versus MGND: $\rho = 0.64 \pm 0.09$, $P = 8 \times 10^{-5}$; Fig. 7a and Supplementary Table 14). In addition, we observed significant excess genetic correlations (FDR < 0.1) for eight disease–subtype and subtype–subtype pairs involving different diseases (Fig. 7a and Supplementary Table 14). Additional secondary analyses are described in the Supplementary Note, Supplementary Fig. 20 and Supplementary Table 15.

Finally, we used the population genetic parameter $F_{ST}$ (refs. 53,54) to quantify genome-wide differences in allele frequency between two subtypes of the same disease. We used discrete subtypes in this analysis. We wished to avoid inferring genetic differences between subtypes that were due to partitions of the cohort that are unrelated to the disease (for example, we expect a nonzero $F_{ST}$ between tall versus short type 2 diabetes cases). Thus, we assessed the statistical significance of nonzero $F_{ST}$ estimates by comparing the observed $F_{ST}$ estimates (between two subtypes of the same disease) to the expected $F_{ST}$ based on matched topic weights (that is, $F_{ST}$ estimates between two sets of healthy controls with topic weight distributions matched to the respective disease subtypes; excess $F_{ST}$; Methods). We determined that 63 of 104 pairs of disease subtypes involving the same disease (spanning 29 of 49 diseases, excluding three diseases that did not have enough controls with matched topic weights) had significant excess $F_{ST}$ estimates (FDR < 0.1; Extended Data Fig. 7 and Supplementary Table 16). For example, the CVD, CER and MGND subtypes of type 2 diabetes had significant excess $F_{ST}$ estimates ($F$-statistic = 0.0003, $P = 0.001$ based on 1,000 matched control sets). This provides further evidence that disease subtypes as determined by comorbidity have different molecular and physiological etiologies. We conclude that disease subtypes defined by distinct topics are genetically heterogeneous.

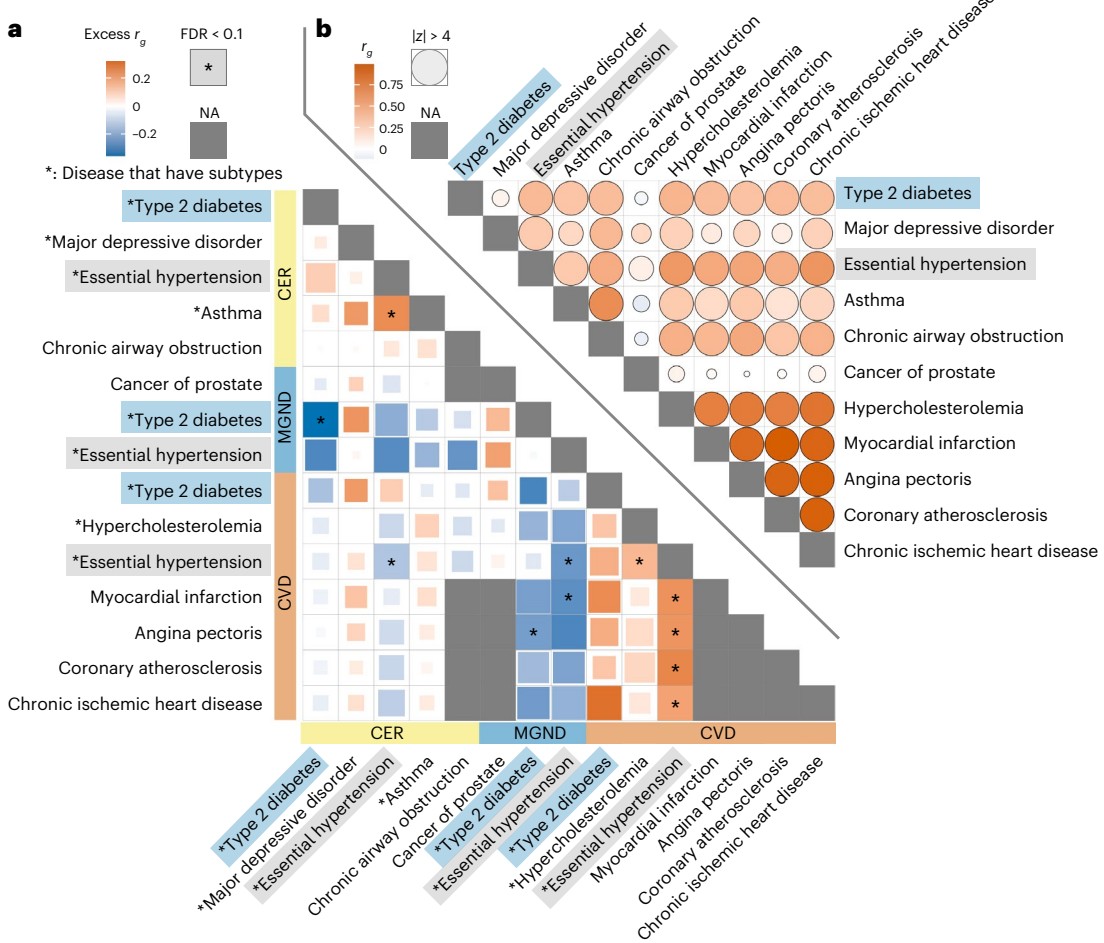

**Fig. 7 | Genetic correlations vary across disease subtypes defined by distinct topics. a**, Excess genetic correlations for pairs of 15 disease subtypes or diseases (nine disease subtypes (denoted with asterisks) + six diseases without subtypes), relative to genetic correlations between the underlying diseases. Full square with asterisk denotes FDR < 0.1; less than full squares have area proportional to z-scores for difference. Gray squares denote NA (pair of diseases without subtypes or pair of the same disease subtype or disease). **b**, Genetic correlations between the underlying diseases. Full circle denotes |z-score| > 4 for nonzero genetic correlation; less than full circles have area proportional to |z-score|. Numerical results are reported in Supplementary Table 14.

## Disease-associated SNPs have subtype-dependent effects

We hypothesized that disease genes and pathways might differentially impact the disease subtypes identified by ATM. We investigated the genetic heterogeneity between disease subtypes at the level of individual disease-associated variants. We used continuous-valued topic weights in this analysis. We used a statistical test that tests for SNP × topic interaction effects on disease phenotype in the presence of separate SNP and topic effects (Methods). We verified via simulations that this statistical test is well calibrated under a broad range of scenarios with no true interaction, including direct effect of topic on disease, direct effect of disease on topic, pleiotropic SNP effects on disease and topic, and nonlinear effects (Supplementary Fig. 21). We also assessed the power to detect true interactions (Supplementary Fig. 22). To limit the number of hypotheses tested, we applied this test to independent SNPs with genome-wide significant main effects on disease (Methods). We thus performed 2,530 statistical tests spanning 888 disease-associated SNPs, 14 diseases and 35 disease subtypes (Supplementary Table 17). We assessed statistical significance using global FDR < 0.1 across the 2,530 statistical tests. We also computed main SNP effects specific to each quartile of topic weights across individuals and tested for different odds ratios in top versus bottom quartiles, as an alternative way to represent SNP × topic interactions; the top/bottom quartile test is more intuitive, but less powerful in most cases.

We identified 43 SNP × topic interactions at FDR < 0.1 (Extended Data Fig. 8, Supplementary Fig. 23 and Supplementary Tables 18 and 19). Here we highlight a series of examples. First, the type 2 diabetes-associated SNP rs1042725 in the *HMGA2* locus has a higher odds ratio in the top quartile of CVD topic weight (1.18 ± 0.02) than in the bottom quartile (1.00 ± 0.02; $P = 3 × 10^{-4}$ for interaction test (FDR = 0.04 < 0.1); $P = 3 × 10^{-7}$ for top/bottom quartile test (FDR = 0.0002 < 0.1)). *HMGA2* is associated with type 2 diabetes[55] and is reported to have functions in cardiac remodeling[56], suggesting that shared pathways underlie the observed SNP × topic interaction. Second, the asthma-associated SNP rs1837253 in the *TSLP* locus has a higher odds ratio in the top quartile of SRD (sense organs, respiratory, dermatologic) topic weight (1.17 ± 0.02) than in the bottom quartile (1.05 ± 0.02; $P = 6 × 10^{-6}$ for interaction test (FDR = 0.004 < 0.1); $P = 1 × 10^{-3}$ for top/bottom quartile test (FDR = 0.08 < 0.1)). *TSLP* has an important role in promoting $T_H2$ cellular responses and is considered a potential therapeutic target, which is consistent with assignment of asthma and atopic/contact dermatitis[57] to the SRD topic (Supplementary Table 4). Two other examples are described in the Supplementary Note. To verify correct calibration, we performed control SNP × topic interaction tests using the same 888 disease-associated SNPs together with random topics that did not correspond to disease subtypes and confirmed that these control tests were well calibrated (Supplementary Fig. 24b). We conclude that genetic heterogeneity between disease subtypes can be detected at the level of individual disease-associated variants.

## Discussion

We have introduced an ATM method to provide a low-rank representation of longitudinal disease records, leveraging age-dependent comorbidity profiles to identify and validate biological subtypes of disease. Our study builds on previous studies on topic modeling[37,38,40,58], genetic subtype identification[13–15] and low-rank modeling of multiple diseases to identify shared genetic components[25–27]. We highlight three specific contributions of our study. First, we incorporated age at diagnosis information into our low-rank representation, complementing the use of age information in other contexts[32,52,59]; we showed that age information is highly informative for our inferred comorbidity profiles in both simulated and empirical data, emphasizing the importance of accounting for age in efforts to classify disease diagnoses. Second, we identified 52 diseases with heterogeneous comorbidity profiles that we used to define disease subtypes, many of which had not previously been identified (Supplementary Table 20); comorbidity-derived disease subtypes were consistent between UK Biobank and All of Us, despite key differences between these cohorts. Third, we used genetic data (including PRS, genetic correlation and $F_{ST}$ analyses) to validate these disease subtypes, confirming that the inferred subtypes reflect true differences in disease etiology.

We emphasize three downstream implications of our findings. First, it is of interest to perform disease subtype-specific genome-wide association studies (GWAS) on the disease subtypes that we have identified here, analogous to GWAS of previously identified disease subtypes[13–15]. Second, our findings motivate efforts to understand the functional biology underlying the disease subtypes that we identified; the recent availability of functional data that are linked to EHR is likely to aid this endeavor[29,60]. Third, the efficient inference of ATM permits identifying age-dependent comorbidity profiles and disease subtypes in much larger EHR datasets[61], although we acknowledge that establishing comprehensive representations of disease topics that are transferable and robust across different healthcare systems and data sources represents a major future challenge.

Our findings reflect a growing understanding of the importance of context, such as age, sex, socioeconomic status and previous medical history, in genetic risk[52,62,63]. To maximize power and ensure accurate calibration, context information needs to be integrated into clinical risk prediction tools that combine genetic information (such as PRSs[1,64]) and nongenetic risk factors. Our work focuses on age, but motivates further investigation of other contexts. We note that aspects of context are themselves influenced by genetic risk factors; hence, there is an open and important challenge in determining how best to combine medical history and/or causal biomarker measurements with genetic risk to predict future events[65].

We note several limitations of our work. First, age at diagnosis information in EHR data may be an imperfect proxy for true age at onset, particularly for less severe diseases that may be detected as secondary diagnoses; although perfectly accurate age at onset information would be ideal, our study shows that that imperfect age at diagnosis information is sufficient to draw meaningful conclusions. Second, raw EHR data may be inaccurate and/or difficult to parse[1]; again, although perfectly accurate EHR data would be ideal, our study shows that imperfect EHR data are sufficient to draw meaningful conclusions. Third, our ATM approach incurs substantial computational cost (Supplementary Table 21); however, analyses of biobank-scale datasets are computationally tractable, with our main analysis requiring only 4.7 h of running time. Additional limitations are described in the Supplementary Note. Despite these limitations, ATM is a powerful approach for identifying age-dependent comorbidity profiles and disease subtypes.

## Online content

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

[1]Big Data Institute, Li Ka Shing Centre for Health Information and Discovery, University of Oxford, Oxford, UK. [2]Department of Statistics, University of Oxford, Oxford, UK. [3]Wellcome Centre for Human Genetics, University of Oxford, Oxford, UK. [4]Department of Epidemiology, Harvard T.H. Chan School of Public Health, Boston, MA, USA. [5]British Heart Foundation Cardiovascular Epidemiology Unit, Department of Public Health and Primary Care, University of Cambridge, Cambridge, UK. [6]Victor Phillip Dahdaleh Heart and Lung Research Institute, University of Cambridge, Cambridge, UK. [7]Program in Medical and Population Genetics, Broad Institute of MIT and Harvard, Cambridge, MA, USA. [8]Chinese Academy of Medical Sciences Oxford Institute, Nuffield Department of Medicine, University of Oxford, Oxford, UK. [9]Department of Radiation Oncology, Peking Union Medical College Hospital, Chinese Academy of Medical Sciences and Peking Union Medical College, Beijing, China. [10]Department of Genetics, Harvard Medical School, Cambridge, MA, USA. [11]Department of Human Evolutionary Biology, Harvard University, Cambridge, MA, USA. [12]Cambridge Baker Systems Genomics Initiative, Department of Public Health and Primary Care, University of Cambridge, Cambridge, UK. [13]Health Data Research UK Cambridge, Wellcome Genome Campus and University of Cambridge, Cambridge, UK. [14]British Heart Foundation Cambridge Centre of Research Excellence, Department of Clinical Medicine, University of Cambridge, Cambridge, UK. [15]Cambridge Baker Systems Genomics Initiative, Baker Heart and Diabetes Institute, Melbourne, Victoria, Australia. [16]The Alan Turing Institute, London, UK. [17]Department of Biostatistics, Harvard T.H. Chan School of Public Health, Boston, MA, USA. [18]These authors contributed equally: Martin Jinye Zhang, Yidong Zhang, Arun Durvasula. [19]These authors jointly supervised this work: Alkes L. Price, Gil McVean. ✉e-mail: xilinjiang@hsph.harvard.edu; aprice@hsph.harvard.edu; gil.mcvean@bdi.ox.ac.uk

## Methods

### Ethics statement

This study analyzed publicly available datasets and hence did not require ethical approval.

### ATM

Our ATM is a Bayesian hierarchical model to infer latent risk profiles for common diseases. The model assumes that each individual possesses several age-evolving disease profiles ('topic loadings'), which summarize the risk over age for multiple diseases that tend to co-occur within an individual's lifetime, namely the age-specific multi-morbidity profiles. At each disease diagnosis, one of the disease profiles is first chosen based on individual weights of profile composition ('topic weights'); the disease is then sampled from this profile conditional on the age of the incidence.

We constructed a Bayesian hierarchical model to infer $K$ latent risk profiles for $D$ distinct common diseases. Each latent risk profile (comorbidity topics) is age-evolving and contains risk trajectories for all $D$ diseases considered. Each individual might have a different number of diseases, while the disease risk is determined by the weighted combination of latent risk topics. We use the following indices:

- $s = 1, \ldots, M$;
- $n = 1, \ldots, N_s$;
- $i = 1, \ldots, K$;
- $j = 1, \ldots, D$;

where $M$ is the number of participants, $N_s$ is the number of diseases within $s^{th}$ participant, $K$ is the number of topics and $D$ is the total number of diseases we are interested in. The plate notation of the generative model is summarized in Extended Data Fig. 1:

- $\theta \in R^{M \times K}$ is the topic weight for all participants (referred to as 'topic weights'), each row of which ($\theta_s \in R^K$) is assumed to be sampled from a Dirichlet distribution with parameter $\alpha$. $\alpha$ is set as a hyperparameter: $\theta_s \sim \mathrm{Dir}(\alpha)$. We used topic weights to assign 'continuous values for disease subtypes' in PRS and SNP × topic analyses.
- $z \in \{1, 2, \ldots, K\}^{\Sigma_s N_s}$ (referred to as 'diagnosis-specific topic probability') is the topic assignment for each diagnosis $w \in \{1, 2, \ldots, D\}^{\Sigma_s N_s}$. The total number of diagnoses across all patients are $\sum_s N_s$. The topic assignment for each diagnosis is generated from a categorical distribution with parameters equal to $s^{th}$ individual topic weight: $z_{sn} \sim \mathrm{Multi}(\theta_s)$. We used diagnosis-specific topic probability to define 'discrete disease subtypes' in excess genetic correlation and excess $F_{ST}$ analyses.
- $\beta(t) \in F(t)^{K \times D}$ is the topic loading that is $K \times D$ functions of age $t$. $F(t)$ is the class of functions of $t$. At each plausible $t$, the following is satisfied: $\sum_j \beta_{ij}(t) = 1$. In practice, we ensure the above is true and add smoothness by constrain $F(t)$ to be a softmax of spline or polynomial functions: $\beta_{ij}(t) = \frac{\exp(p_{ij}^T \phi(t))}{\sum_{j=1}^{D} \exp(p_{ij}^T \phi(t))}$, where $p_{ij}^T \phi(t)$ is polynomial and spline functions of $t$; $p_{ij} = \{p_{ijd}\}; d = 1, 2, \ldots, P$; $P$ is the degree of freedom that controls the smoothness; $\phi(t)$ is polynomial and spline basis for age $t$.
- $w \in \{1, 2, \ldots, D\}^{\Sigma_s N_s}$ are observed disease diagnoses. The $n^{th}$ diagnosis of $s^{th}$ participant $w_{sn}$ is sampled from the topic $\beta_{z_{sn}}(t) \in F(t)^D$ (chosen by $z_{sn}$): $w_{sn} \sim \mathrm{Multi}(\beta_{z_{sn}}(t_{sn}))$, where $t_{sn}$ is the the observed age at diagnosis of $w_{sn}$.

The variables of interest are disease topics $\beta(t)$, individual (patient)-level topic weight $\theta$ and diagnosis-specific topic probability $z$. The innovative element in our model is age-evolving risk profiles, which are achieved by modeling the comorbidity trajectories $\beta(t) \in F(t)^{K \times D}$ as functions of age. We parameterized functionals $F(t)$ as linear, quadratic, cubic polynomials, and cubic splines with one, two and three knots. We use prediction odds ratio to decide the optimal model structure

including the function forms and the number of topics; we use evidence lower bound (ELBO) to choose the optimal inference results (with random parameter initialization) for the same model structure (Supplementary Table 1).

### Inference of ATM

The variables of interest are global topic parameter $\beta(t)$, individual (patient)-level topic weight $\theta$ and diagnosis-specific topic probability $z$ of each diagnosis. We adopt an EM strategy, where in the E-step we estimate posterior distribution of $\theta$ and $z$, and then in the M-step we estimate $\beta$ that maximizes the ELBO. For the E-step, we used a collapsed variational inference; for the M-step, we used local variational inference (details are described in the Supplementary Note).

We extract topic weights at the patient level and diagnosis level from the posterior distribution $q(z)$, which is a categorical distribution (equation 8 of Supplementary Note). Our model has the desired property that both patients and diagnoses are assigned to comorbidity topics. We listed the following definitions in the paper that are derived from $q(z)$:

- Each diagnosis has a 'diagnosis-specific topic probability', which is computed as $E_q\{z_n\}$.
- Each patient has a posterior 'topic weight' $\theta_s$, which is a Dirichlet distribution $\theta_s \sim \mathrm{Dir}(a + \sum_{n=1}^{N_s} E_q\{z_n\})$. The topic weights of each patient are defined as the mode of this Dirichlet distribution $\frac{\sum_{n=1}^{N_s} E_q\{z_n\}}{\sum_{i=1}^{K} \sum_{n=1}^{N_s} E_q\{z_{ni}\}}$ (we used $\alpha = 1$, which puts a noninformative prior on the topic weights). Topic weight is the low-rank representation of disease history, which is used in excess PRS and SNP × topic interaction analyses.
- The average topic assignments of disease $j$ are the mean overall incidences $E_q\{z_{sn \in \{w_{sn}=j\}}\}$. This metric is used to measure which comorbidity topic a disease is associated with (Fig. 4b), and it is equivalent to a weighted average of topic loadings (equation 5 in Supplementary Note shows the link between diagnosis-specific topic probability and topic loading). A disease assigned to multiple topics is considered to have comorbidity subtypes.
- A hard assignment of a patient diagnosis to a comorbidity-derived subtype is based on the maximum value of the vector $E_q\{z_n\}$. The incident disease is assigned to topic $\mathrm{argmax}_i(E_q\{z_{ni}\})$.

### Metrics for evaluating ATM

ATM is evaluated for different purposes, which require different metrics (Supplementary Table 1). Here we list the details of the four metrics considered: prediction odds ratio, evidence lower bound (ELBO), the area under the precision-recall curve (AUPRC)[66] and co-occurrence odds ratio.

**Prediction odds ratio.** We used prediction odds ratio to compare models of different topic numbers and configuration of age profiles. Briefly, prediction odds ratio is defined on 20% held-out test data as the odds that the true diseases are within the top 1% of diseases predicted by ATM (trained on 80% of the training set and uses earlier diagnoses as input), divided by the odds that the true diseases are within the top 1% of diseases ranked by prevalence. Details of computing prediction odds ratio for each patient are described in the Supplementary Note.

**Evidence lower bound (ELBO).** ELBO evaluated the accuracy of the variational inference method on a specific dataset[39]. The mathematical expression of ELBO for ATM is equation 9 in the Supplementary Note. We computed ELBO when fitting ATM to UK Biobank with 19 choices of the number of topics (5–20, 25, 30 and 50) and six choices of age profile configuration (linear, quadratic polynomial, cubic polynomial and splines with one, two and three knots). Each model is run ten times

with random initializations. We choose the model that has the highest ELBO after converging.

**AURPC.** To evaluate whether a model could capture the comorbidity subtypes in simulation analysis, we compute the precision, recall and AUPRC to correctly assign disease diagnosis to the true topic. The topic of each diagnosis is determined by diagnosis-specific topic probability. Note we could only evaluate AUPRC in simulations where the truth is known.

**Co-occurrence odds ratio.** To verify that the comorbidity profiles are capturing diseases that are more likely to co-occur within the same individual, we estimate the odds ratio of the disease duo, trio, quartet and quintet that are captured by the topic versus that of random combinations. Details of computing co-occurrence odds ratio for each model are described in the Supplementary Note.

### Simulations of ATM method
To test whether the algorithm could assign disease diagnosis to correct comorbidity profiles, we simulated diseases using the following parameters:

- $M = 10,000$;
- $\bar{N}_s = 6.1$;
- $N_s \sim \exp\{\bar{N}_s\}$;
- $D = 20$;
- $K = 2$;

where $M$ is the number of individuals in the population, $\bar{N}_s$ is the average number of diseases for each individual, $D$ is the total number of diseases and $K$ is the number of comorbidity topics. The distribution of disease number per-individual $N_s$ is sampled from an exponential distribution, which matches those from UK Biobank data (Supplementary Fig. 25). According to equation 3.1 in ref. 67, whether the topic model could capture the true latent structure is determined by the signal-to-noise ratio and could be evaluated with limits $M \to \infty; D \to \infty; \frac{D}{M} \to \delta$, where $\delta$ is a constant. Therefore, we choose $D$ and $M$ that make $\frac{D}{M}$ similar to those of the UK Biobank dataset (samples size = 282,957; distinct disease number = 349).

Details of topic loadings (Supplementary Fig. 26) and topic weights used in simulations are described in the Supplementary Note.

We simulated diseases with distinct comorbidity subtypes by combining diseases from distinct topics and labeling them as a single disease, using the parameters described above. We consider the following two scenarios: (1) the subtypes of diseases have the same age at diagnosis distribution; (2) the subtypes of disease have distinct age at diagnosis distribution. We first chose one disease (disease A) and then sampled a proportion of a second disease (disease B) to label as disease A. The proportion is varied to create a different sample size ratio of the two subtypes. In scenario one, disease B is a disease that has the exact same age distribution as disease A but from the other topic. In scenario two, disease B is from the other topic and has a different age distribution (age at diagnosis moves up for 20 years, 10 years or 5 years, respectively) than disease A. After changing the labels of disease B to be the same as disease A, we use ATM/LDA to infer diagnosis-specific topic probability to assign diagnoses to the topics.

To evaluate whether a model could capture the comorbidity subtypes, we compute the precision, recall, and AUPRC of correctly classifying incident disease B to be from the correct topic. The topic of each diagnosis is determined by diagnosis-specific topic probability. We use other diseases from the same topic of disease B to benchmark the topic label. Topic modeling on the simulated data is performed with both ATM and LDA (both implemented using collapsed variational inference for fair comparison) to compare the performances.

We evaluate the subtype classification with varying values of the following four parameters: ratio of sample sizes between the two

subtypes, simulated population size, number of distinct diseases and difference of age distribution (details are described in the Supplementary Note).

### UK Biobank comorbidity data
We analyzed comorbidity data from 282,957 UK Biobank samples with diagnoses for at least two of the 348 focal diseases that we studied (see next paragraph). We use the hospital episode statistics (HES) data within the UK Biobank dataset, which uses the ICD-10/ICD-10CM coding system; the average record span of HES data is 28.6 years. Codes starting with letters from A to N are kept as they correspond to disease codes (as opposed to procedure codes). The disease records were mapped from ICD-10/ICD-10-CM codes to Phecodes using a three-step procedure (details are described in the Supplementary Note), and the method is implemented in ATM software.

The mapped Phecodes are filtered to keep only the first diagnosis for the recurrent diseases within a patient. The age at diagnosis for each disease is computed as the difference between the month of birth to the episode starting date. We then computed the occurrence of each disease in the UK Biobank and kept 348 that have more than 1,000 occurrences (Supplementary Table 4). Starting with all 488,377 UK Biobank participants (including both European and non-European ancestries), we filtered the patients to keep only those who have at least two distinct diseases from the 348 focal diseases, as we are most interested in the comorbidity information. We treated death as an additional disease (8,666 records) to evaluate if certain comorbidities are more likely to lead to fatal events. The procedure leaves us 1,726,144 distinct records across 282,957 patients.

To name the topics inferred from the UK Biobank, we take the sum of 'average topic assignments' (Inference of ATM) over diseases for each Phecode system and extract the three most common Phecode disease systems. The following six topics are named using the three most common Phecode disease systems: NRI (neoplasms, respiratory, infectious diseases), CER (cardiovascular, endocrine/metabolic, respiratory), SRD (sense organs, respiratory, dermatologic), FGND (female genitourinary, neoplasms, digestive), MGND (male genitourinary, neoplasmas, digestive) and MDS (musculoskeletal, digestive, symptoms). For four topics that are predominantly associated with one system, we name them based on their top associated Phecode system: LGI (lower gastrointestinal), UGI (upper gastrointestinal), CVD (cardiovascular) and ARP (arthropathy).

We present focal diseases by selecting diseases with the highest average topic loading between ages 30 and 81 years. We chose the top seven diseases for visualization, as we found more diseases would be harder to read on a plot.

To compare the comorbidity heterogeneity between age groups, we group the incidences for each disease to the following two age groups: young group (<60 years of age) and old group (≥60 years of age). We compute the average topic assignment of each group as described in the section 'Inference of ATM.' Additionally, we inferred topics for male (984,554 records in 156,366 individuals) and female (741,590 records in 126,591 individuals) populations, respectively, using ATM with ten topics and spline function with one knot. We extract the average topic assignment for each disease and use Pearson's correlation to match the topics for both sexes to the topics inferred on the entire population.

We assigned diagnoses to discrete subtypes using maximum diagnosis-specific topic probability. We focus our genetic heterogeneity analysis on 52 diseases that have at least 500 incidences assigned to a secondary topic.

### All of Us comorbidity data
We analyzed EHR data collected in the EHR domain of All of Us samples, which includes both primary care and secondary care data. The average distance between the first and last diagnoses is 7.9 years (versus

7.0 years in UK Biobank); the average record span period is unknown, but we hypothesized that it is likely to be considerably larger than 7.9 years (versus 28.6 years in UK Biobank). Disease codes in the All of Us EHR domain are coded in SNOMED CT. Details of mapping from SNOMED CT to Phecode are described in the Supplementary Note and implemented in ATM software. We kept 233 Phecodes that overlap with the 348 diseases analyzed in the UK Biobank. We kept the first diagnosis for recurrent diseases in each patient. After mapping, we are left with 3,098,771 diagnoses spanning 211,908 All of Us samples. We run ATM with topic numbers from 5 to 20 and age functions configured as splines with two knots (degree of freedom = 5) on the All of Us comorbidity data and computed prediction odds ratio (using fivefold cross-validation) and ELBO (on all 211,908 samples).

## Comparing disease topics between UK Biobank and All of Us
We compared the optimal models from UK Biobank (ten topics, degree of freedom = 5) and All of Us (13 topics, degree of freedom = 5). We constrained our analyses on 233 of the 348 diseases that are shared between the two datasets. We performed three analyses to compare the comorbidity patterns from the two datasets.

First, we computed the correlation of topic loadings from two datasets. Because the topic loadings are functions of age, we computed their correlations using four different ways to summarize age information—topic loadings averaged across age and topic loadings at ages 50, 60 and 70 years. For each UK Biobank topic, we found its most similar All of Us topic that has a maximum correlation of topic loadings (averaged across age).

Second, we computed the cross-population prediction odds ratio, using the All of Us topics to predict comorbidity patterns in UK Biobank data. We divided the UK Biobank samples into ten jackknife blocks and computed prediction odds ratios on each leave-one-out sample.

Third, we compared the correlation of comorbidity profiles (measured by average topic assignments; see Methods for definition) for 233 diseases that are shared between the two populations. We define correlations between topic assignments as the correlation between UK Biobank average topic assignments and All of Us average topic assignments mapped to UK Biobank topic space (details are described in the Supplementary Note).

## UK Biobank genotype data
For genetic correlation analysis, $F_{ST}$ and SNP × topic interaction analyses, we used genetic data from 488,377 UK Biobank participants (before restricting to 282,957 samples with at least two of the 348 diseases studied). For PRS and heritability estimation of the ten topics, we constrained our analysis to 409,694 British Isle ancestry individuals to adjust for population structure. We choose different sets of SNPs that are practical for each method (details are described in the Supplementary Note).

## Polygenic risk scores analysis
Although population stratification cannot be excluded[68], to adjust for and minimize the impact of population stratification, we applied mixed-effect association model to samples of the British Isle ancestry group ($n = 409,694$) to compute PRS for ten heritable diseases that have the highest heritability $z$-scores. We used BOLT-LMM to construct genome-wide PRS[50]. Details of creating balanced case–control and SNP selection are described in the Supplementary Note. For each disease, we used fivefold cross-validation to estimate effect sizes using BOLT-LMM and computed the PRS on the held-out testing set. We used linear regression between continuous-valued topic weights and PRS to compute the excess PRS over different topics.

We compute the subtype-specific relative risk for each percentile of PRS using the following formula:

$$\mathrm{RR}_{\mathrm{pt},s} = \frac{n_{\mathrm{pt},s} \times 100}{n_s},$$

where $\mathrm{RR}_{\mathrm{pt},s}$ is the relative risk of $s$ subtype for the $pt^{\mathrm{th}}$ PRS percentile (computed for the entire population), $n_{\mathrm{pt},s}$ is the number of cases in $s$ subtype that has PRS within the $pt^{\mathrm{th}}$ percentile and $n_s$ is the number of cases in the $s$ subtype.

## Genetic correlation analysis
We used discrete subtypes in genetic correlation analysis (different from the PRS analysis above). For each disease and disease subtype, we used a case–control matching strategy to construct data to estimate coefficients for genetic correlation analysis. We used a one-to-four case–control ratio, matching sex, BMI, year of birth and 40 genetic principal components. We used PLINK 1.9 (ref. [69]) to perform logistic regression with sex and the top ten principal components as covariates. We used linkage disequilibrium score regression (LDSC)[2] and summary statistics from the logistic regression to estimate the heritability for each disease or disease subtype that has more than 1,000 incidences (378 = 30 disease subtypes + 348 diseases). We focus on 71 diseases and 18 disease subtypes of the 378 disease subtypes and diseases that have heritability $z$-score above 4 for genetic correlation analysis.

We used LDSC and summary statistics from the logistic regression to compute genetic correlation for each pair of disease–disease, disease–subtype and subtype–subtype. We report the estimate of genetic correlation and $z$-scores. Additionally, for pairs that involve subtypes (disease–subtype or subtype–subtype), we compute the excess genetic correlation, defined as the difference between the genetic correlation involving subtypes (disease–subtype and subtype–subtype) and the genetic correlation involving all disease diagnoses (disease–disease). For example, the genetic correlation between type 2 diabetes–CER and hypertension–CVD is compared to the genetic correlation between all type 2 diabetes and all hypertension. We note that genetic correlations between subtypes of the same disease are compared to 1. We only reported $P$-values of excess genetic correlation when both genetic correlation estimation has s.e. < 0.1 and at least one of the genetic correlations has |$z$-score| > 4.

To avoid potential collider effects where subtypes are defined by topic components that are independent of the diseases, we performed the same genetic correlation analyses but match cases in each subtype with controls with similar topic loadings (details are described in the Supplementary Note).

## $F_{ST}$ analysis
We used discrete subtypes in $F_{ST}$ analysis (same as genetic correlation analysis above; different from the PRS analysis). To evaluate the genetic heterogeneity between disease subtypes, we estimated the $F_{ST}$ across subtypes for 52 diseases that have at least 500 incidences assigned to a secondary topic. To test the statistical significance of $F_{ST}$, we adopted a permutation strategy by sampling controls with matched topic weights and sample size for each disease subtype and computed $F_{ST}$ across the subtype-matched control groups (details are described in the Supplementary Note). The $F_{ST}$s are computed using PLINK 1.9's weighted mean across all genotyped SNPs.

We obtained 1,000 permutation samples and reported the permutation $P$-value for each disease. Under the assumption that causal and noncausal variants have similar allele frequency differences across the subtypes, $F_{ST}$ is a measure of causal genetic effect heterogeneity across subtypes.

## SNP × topic interaction test
We used continuous-valued topic weights in the SNP × topic interaction analysis (same as the PRS analysis; different from the genetic correlation and $F_{ST}$ analyses). We focus on 14 diseases that have heritability $z$-score above 4. We fit following a logistic regression model:

$$\mathrm{logit}(p) = \beta_0 + \beta_1 \times T + \beta_2 \times T^2 + \beta_3 \times G + \beta_4 \times G \times T,$$

where $T$ is individual topic weights for a specified topic, $G$ is the genotype and $p$ is the probability of getting the disease. We computed the test statistics under the null that $\beta_4 = 0$.

Simulations (below) showed that the interaction test is underpowered when the variant effects are small; we focused on the set of GWAS SNPs that reach genome-wide significance level. We used LD-clumping at $R^2 > 0.6$ to keep moderately independent variants. We computed the test statistics using the model above (for testing $\beta_4 = 0$) and computed study-wise FDR across 2,530 disease–topic pairs. We used QQ plots to check that interaction test statistics computed using all nonsubtype topics for each disease (which are expected to be null) were well calibrated (Supplementary Fig. 24b).

As an alternative way to verify the interactions, we divided cases into quartiles based on topic weights (which define disease subtypes continuously) for each disease–topic pair and randomly sampled two controls that match the topic weights for each case. We estimated the main effect sizes for all GWAS SNPs within each quartile of topic weight and compared the effects between the top and bottom quartiles of topic weights. For visualization, we used GWAS SNPs that have no interaction effect (above, $P > 0.05$) as background SNPs.

### Simulations of SNP × topic interaction
We simulated comorbidity with genetics to test the interaction between genetic and comorbidity topics. We simulated 100 independent variants with minor allele frequency (MAF) randomly sampled from the MAF of 888 independent disease-associated SNPs. We assumed an additive model and simulated genotypes for the population using Hardy–Weinberg equilibrium. We simulated three types of genetic effects on topic and diseases, based on the simulation framework described in the Simulations of ATM method (genetics–topic, genetic–disease–topic and genetic–topic interaction; details are described in the Supplementary Note).

We simulated with varying disease–topic or topic–disease causal effects with 50 repetitions at each causal effect size. The simulated data are fed to the ATM to infer the topic weights for interaction testing.

### Reporting summary
Further information on research design is available in the Nature Portfolio Reporting Summary linked to this article.

### Data availability
UK Biobank data are publicly available at https://www.ukbiobank.ac.uk/ (application number 12788). All of Us data are publicly available at https://allofus.nih.gov. LD-scores and HAPMAP3 SNP list are available at https://data.broadinstitute.org/alkesgroup/LDSCORE.

### Code availability
Open-source software implementing the ATM method is available at https://github.com/Xilin-Jiang/ATM. BOLT-LMM 2.3 is available at https://alkesgroup.broadinstitute.org/BOLT-LMM/. Heritability and genetic correlation analysis were performed using LDSC, which is available at https://github.com/bulik/ldsc. PLINK v1.9, which was used for $F_{ST}$ and association tests, is available at https://www.cog-genomics.org/plink/. All codes generated in this study are available at https://zenodo.org/record/8304651 (https://doi.org/10.5281/zenodo.8304651).

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

## Acknowledgements
This research has been conducted using the UK Biobank Resource (application 12788). The All of Us Research Program is supported by the National Institutes of Health, Office of the Director: Regional Medical Centers (1 OT2 OD026549, 1 OT2 OD026554, 1 OT2 OD026557, 1 OT2 OD026556, 1 OT2 OD026550, 1 OT2 OD 026552, 1 OT2 OD026553, 1 OT2 OD026548, 1 OT2 OD026551, 1 OT2 OD026555 and IAA: AOD 16037), Federally Qualified Health Centers (HHSN 263201600085U), Data and Research Center (5 U2C OD023196), Biobank (1 U24 OD023121), The Participant Center: (U24 OD023176), Participant Technology Systems Center (1 U24 OD023163), Communications and Engagement (3 OT2 OD023205 and 3 OT2 OD023206) and Community Partners (1 OT2 OD025277, 3 OT2 OD025315, 1 OT2 OD025337 and 1 OT2 OD025276). In addition, the All of Us Research Program would not be possible without the partnership of its participants. This work was funded by Wellcome (215096/Z/18/Z to X.J. and 100956/Z/13/Z to G.M.; https://wellcome.org); the Li Ka Shing Foundation (to G.M.; https://www.lksf.org); NIH (grants R01 HG006399, R01 MH101244 and R37 MH107649 to A.L.P.); the Alan Turing Institute (https://www.turing.ac.uk), Health Data Research UK (https://www.hdruk.ac.uk), the Medical Research Council UK (https://mrc.ukri.org), the Engineering and Physical Sciences Research Council (EPSRC; https://epsrc.ukri.org) through the Bayes4Health program (grant EP/R018561/1) and AI for Science and Government UK Research and Innovation (UKRI (to C.H.); https://www.turing.ac.uk/research/asg); British Heart Foundation award reference number CH/12/2/29428 (to X.J.); Munz Chair of Cardiovascular Prediction and Prevention and the NIHR Cambridge Biomedical Research Center (BRC-1215-20014; NIHR203312) and UK Economic and Social Research 878 Council (ES/T013192/1 to M.I.). This work was supported by core funding from the British Heart Foundation (RG/13/13/30194 and RG/18/13/33946), Cambridge BHF Center of Research Excellence (RE/18/1/34212) and NIHR Cambridge Biomedical Research Center (BRC-1215-20014 and NIHR203312). The funders had no role in study design, data collection and analysis, decision to publish or preparation of the manuscript. This work uses data provided by patients and collected by the NHS as part of their care and Support. Computation used the Oxford Biomedical Research Computing (BMRC) facility, a joint development between the Wellcome Center for Human Genetics and the Big Data Institute supported by Health Data Research UK and the NIHR Oxford Biomedical Research Center. The views expressed are those of the authors and not necessarily those of the NHS, the NIHR or the Department of Health and Social Care. We thank K. Dey (Sloan Kettering Institute), L. Kelly (University of Oxford) and Yunlong Jiao (University of Oxford) for the helpful discussion.

## Author contributions
X.J., C.H., A.L.P. and G.M. designed the study. X.J. implemented the software and visualized the results. X.J., Y.Z. and A.D. analyzed the data. X.J., A.L.P. and G.M. wrote the manuscript with assistance from M.J.Z., Y.Z., A.D., M.I. and C.H.

## Competing interests
G.M. is a director of and shareholder in Genomics PLC and is a partner in Peptide Groove LLP. The other authors declare no competing financial interests.

## Additional information
**Extended data** is available for this paper at https://doi.org/10.1038/s41588-023-01522-8.

**Correspondence and requests for materials** should be addressed to Xilin Jiang, Alkes L. Price or Gil McVean.

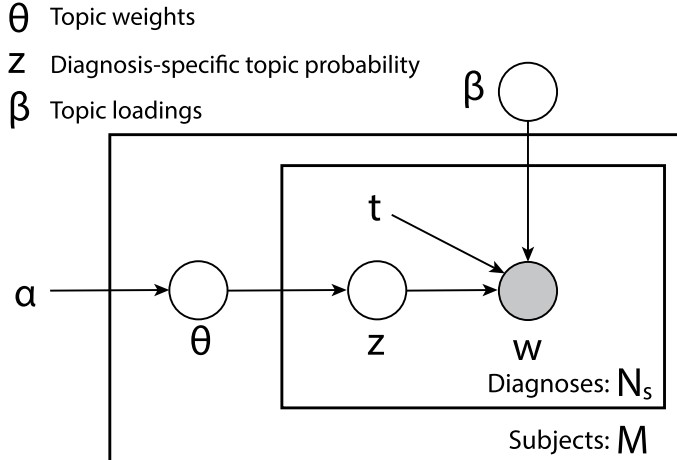

**Extended Data Fig. 1 | Plate notation of ATM generative model.** *M* is the number of subjects; $N_s$ is the number of records within $s^{th}$ subject. All plates (circles) are variables in the generative process, where the plates with shade *w* is the observed variable and plates without shade are unobserved variables to be inferred; β is the topic weight; *z* is diagnosis-specific topic probability; *t* is the age at diagnosis for each diagnosis; β is the topic loadings, which are functions of age *t*; α is the (non-informative) hyperparameter of the prior distribution of θ. The generative process is described in the Methods and Supplementary Note.

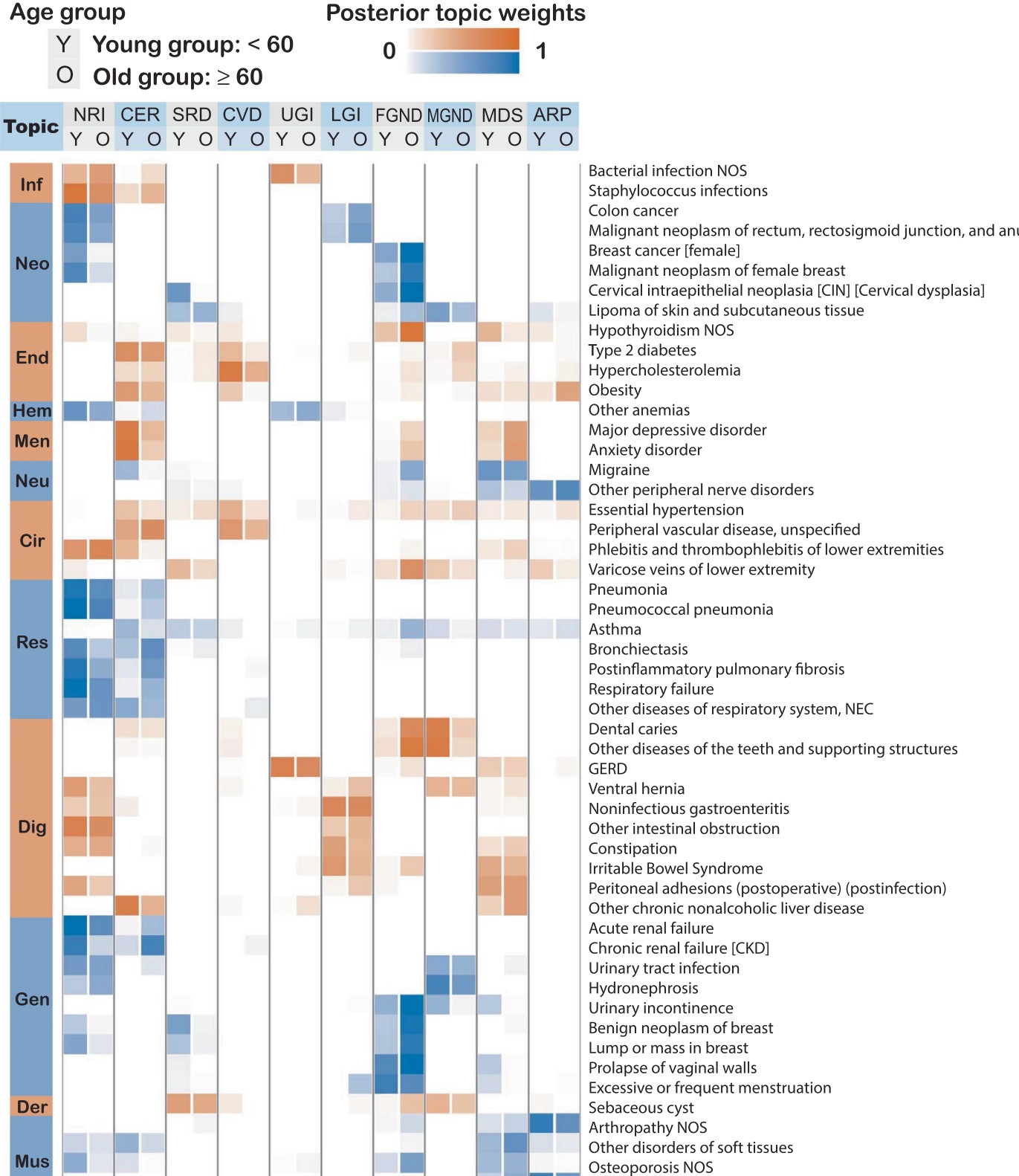

**Extended Data Fig. 2 | Posterior topic distributions are different between age groups for diseases that have subtypes.** The figure has the same legends as Fig. 3 but focusing on 52 diseases that have a subtype with at least 500 incidences. Distribution of average topic assignment for these 52 diseases is reported in Supplementary Fig. 15.

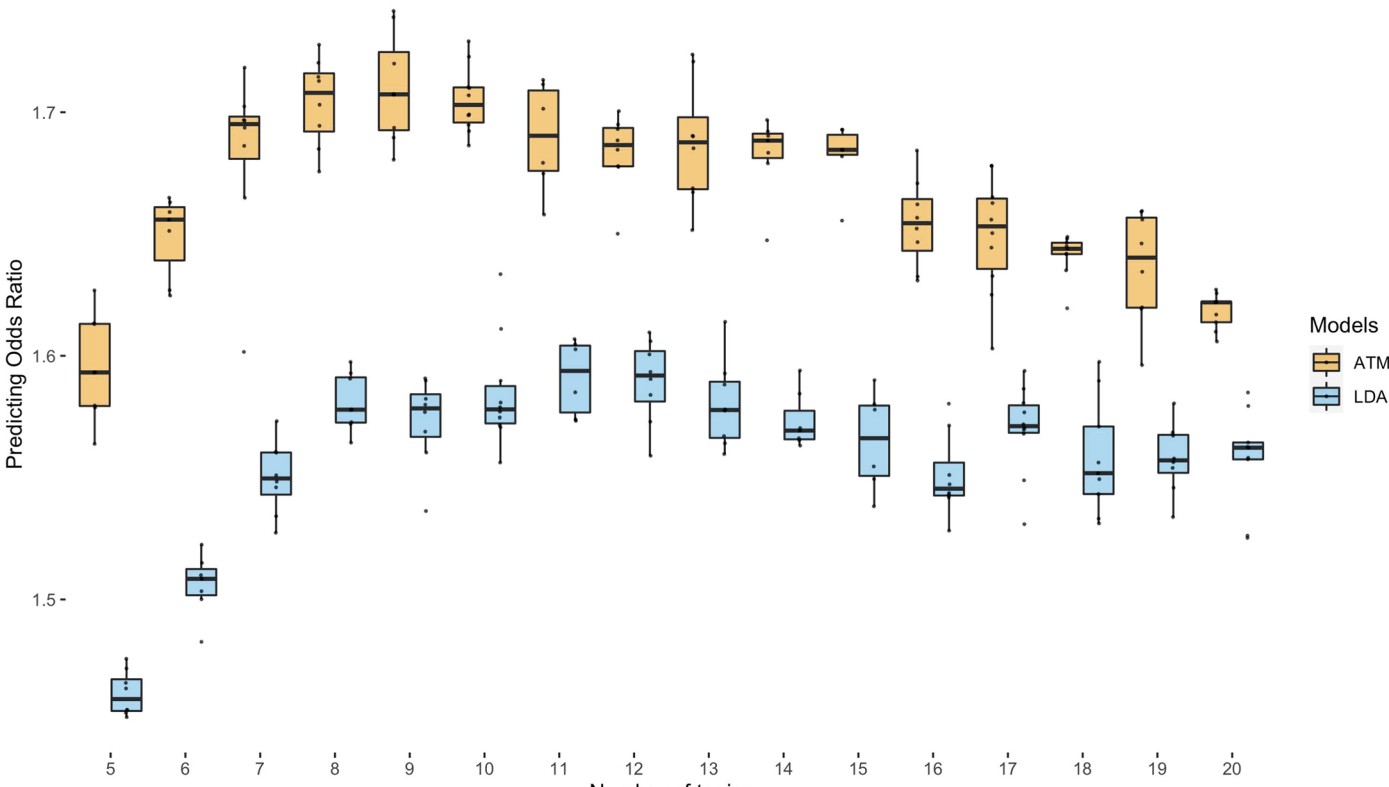

**Extended Data Fig. 3 | Comparison of prediction odds ratio between LDA and ATM.** Each dot represents results from running either ATM or LDA on the same random training and testing split. The models were run with different topic numbers, and we chose a cubic spline with one knot for configuring ATM topic loadings. The prediction odds ratios are computed on the testing data using topic loadings inferred from the training data and topic weights inferred using previous diseases of testing individuals. The odds ratios are between the odds that target diseases are within model-predicted top percentile disease set versus the odds that target diseases are within the prevalence-ordered top percentile disease set. For the optimal model with 10 topics, ATM has an average prediction odds ratio 1.71 (across 10 random training-testing splits); LDA has an average prediction odds ratio 1.58 (across 10 random training-testing splits). Box plots show the distributions of the dots; center, box bounds, and whisker ends denote median, quartiles, and minima/maxima.

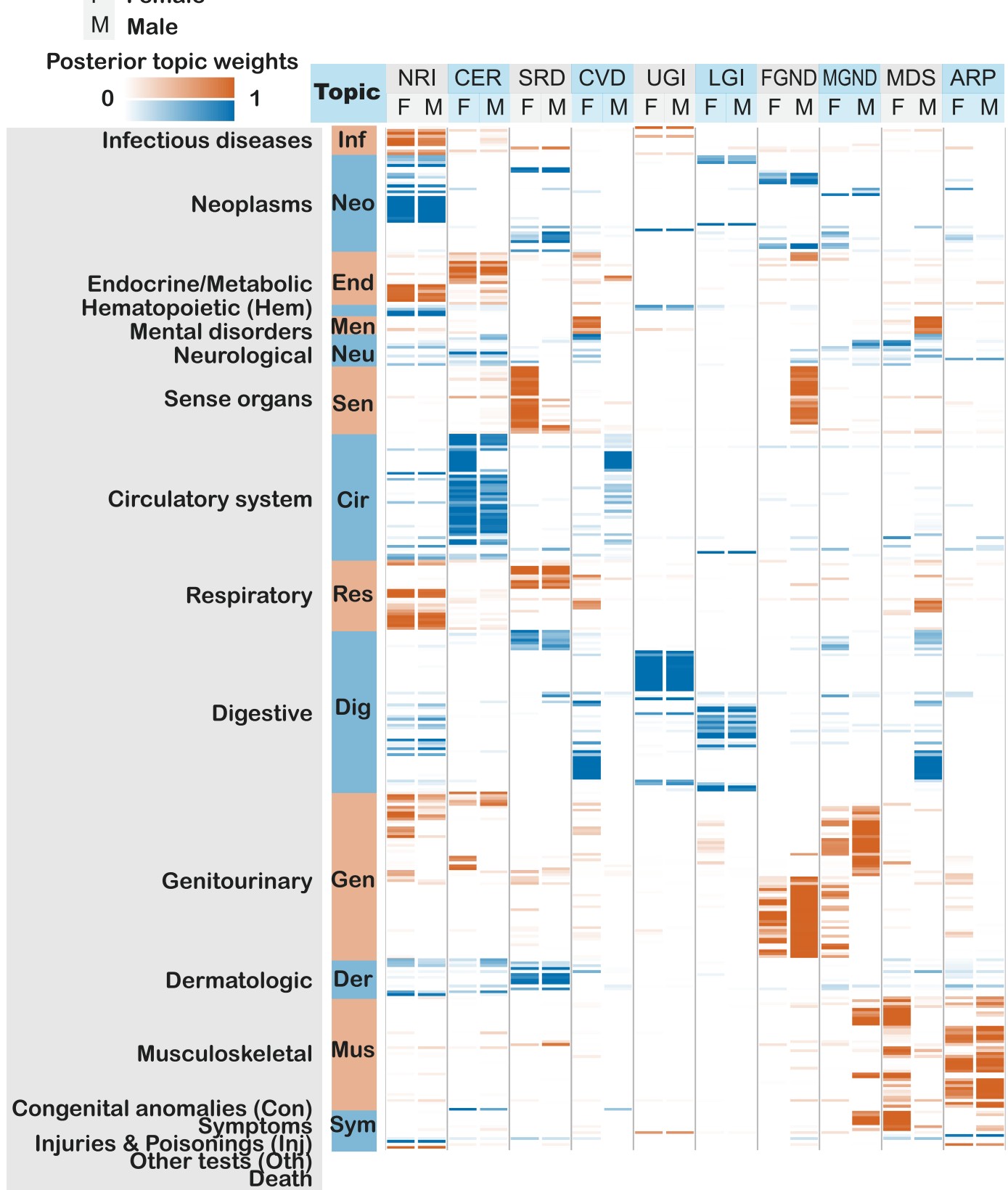

**Extended Data Fig. 4 | Posterior topic distributions of female and male populations.** The figure is the same as Fig. 3 but comparing the topics that are inferred from female and male populations separately.

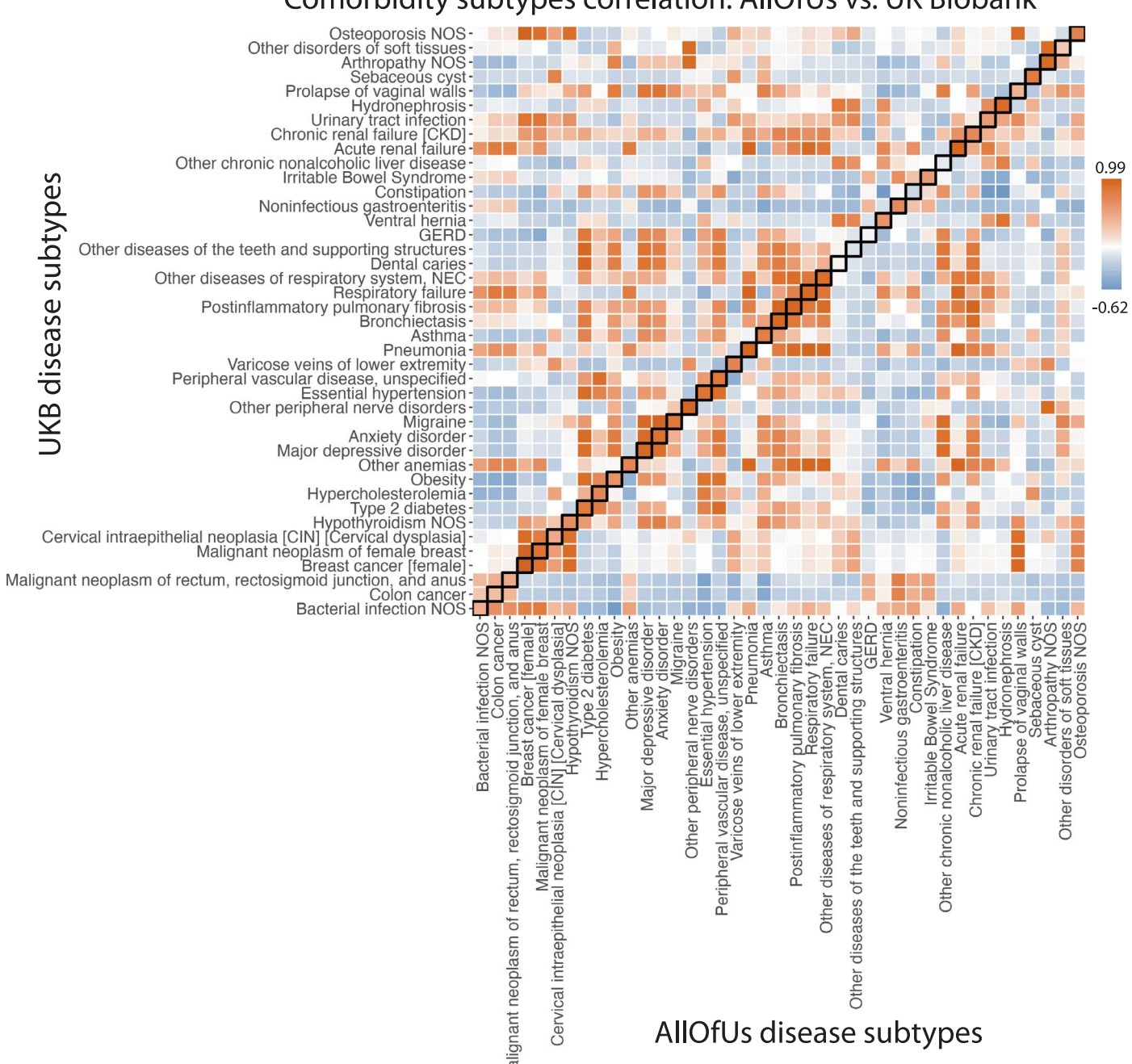

**Extended Data Fig. 5 | Subtype correlations between UK Biobank and All of Us for 41 diseases that are presented in both datasets and have subtypes in UK Biobank.** Each box of the heatmap shows the correlation of average diagnosis-specific topic probability between a disease from All of Us and the other disease from UK Biobank. The diagnosis-specific topic probabilities from All of Us were mapped to UK Biobank based on proportional variance between the two topic spaces (Methods).

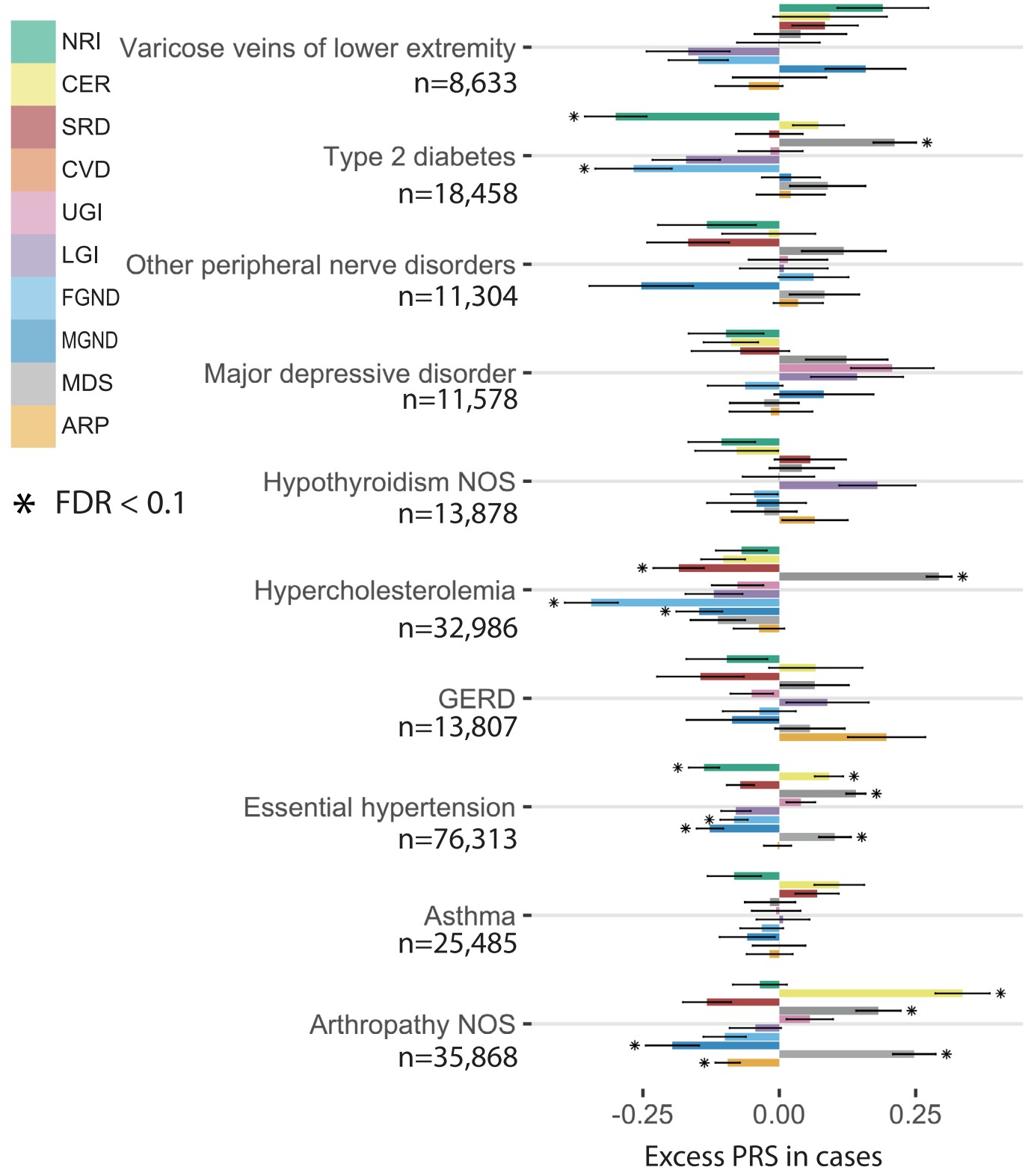

**Extended Data Fig. 6 | Excess PRS analysis for all topics across 10 diseases (selected by heritability z-score).** The bar plot shows the estimated changes in s.d. of PRS per unit changes in the patient topic weight in disease cases. The PRS is estimated using all the cases in British Isle Ancestry. Error bars denote the 95% confidence interval. The stars show disease-topic pairs that are significant at FDR = 0.05. Numerical results are reported in Supplementary Table 13.

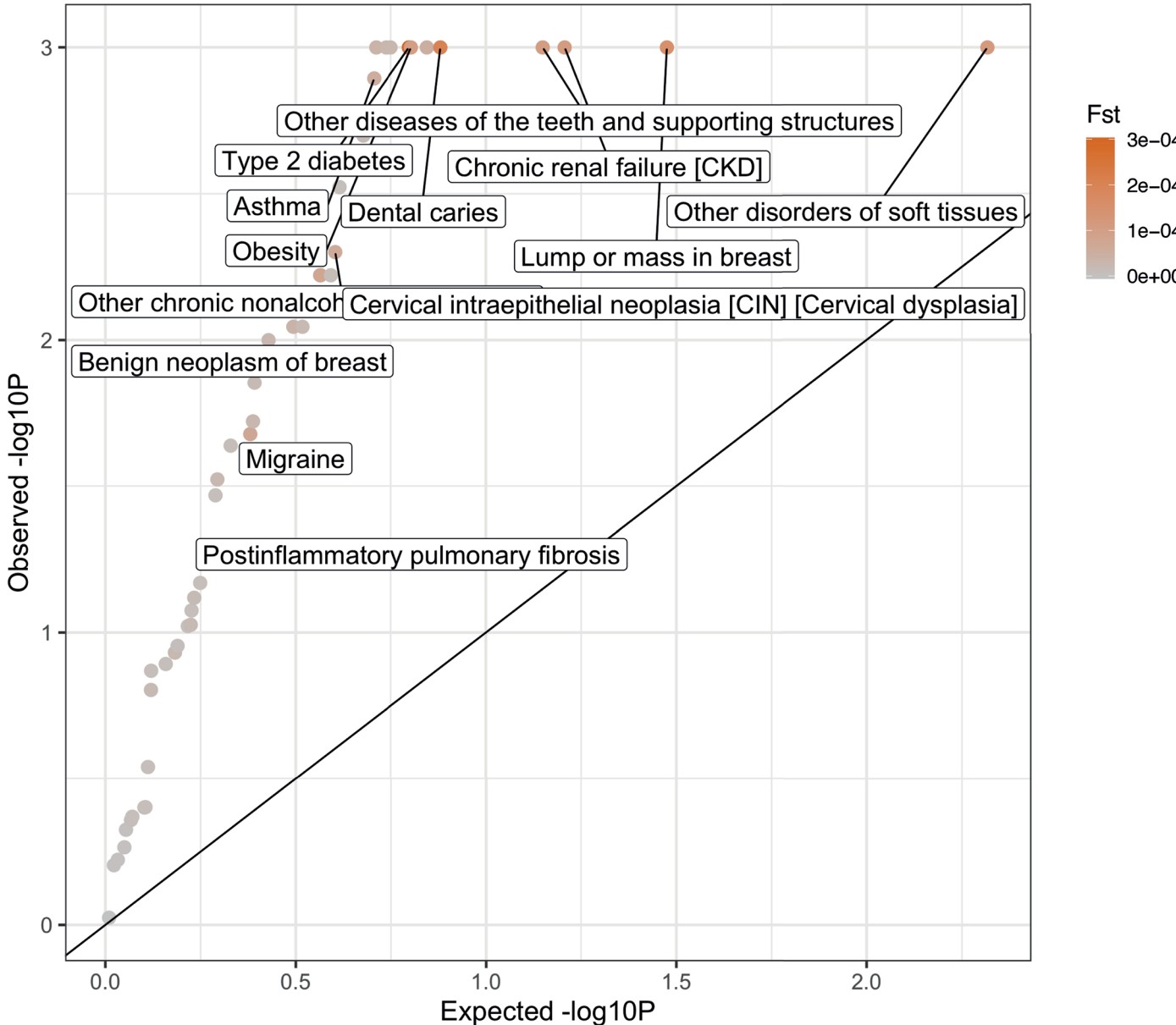

**Extended Data Fig. 7 | Excess $F_{ST}$ of disease subtypes compared with controls with matched topic weights.** *P*-values are for testing case-$F_{ST}$ significantly higher than control-$F_{ST}$ of similar topic weight distribution. The permutation controls are sampled for 1,000 times with the same topic weights distribution and sample size to the disease subtypes. We focus on 49 of the 52 diseases that have more than one subgroup of at least 500 cases. Subtypes are defined based on the maximum value of the diagnosis-specific topic probability. Three diseases ('hypertension', 'hypercholesterolemia', and 'arthropathy') are excluded as there are not enough controls that match the topic weights of cases. The color shows the value of $F_{ST}$ across subtypes. Exact *P*-values are reported in Supplementary Table 16.

A · rs1042725 x CVD for Type 2 diabetes
chr12: 12q14.3 (nearest gene: *HMGA2*)
P = 3e-04 for interaction
P = 3e-07 for top/bottom difference
N = 73,239

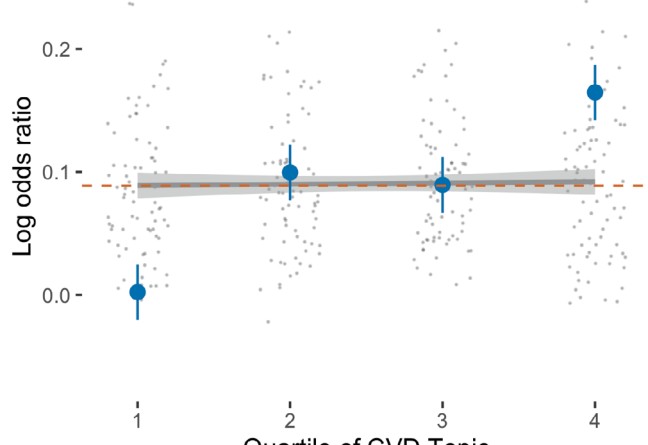

B · rs1837253 x SRD for Asthma
chr5: 5q22.1 (nearest gene: *BCLAF1P1 - TSLP*)
P = 6e-06 for interaction
P = 1e-03 for top/bottom difference
N = 94,284

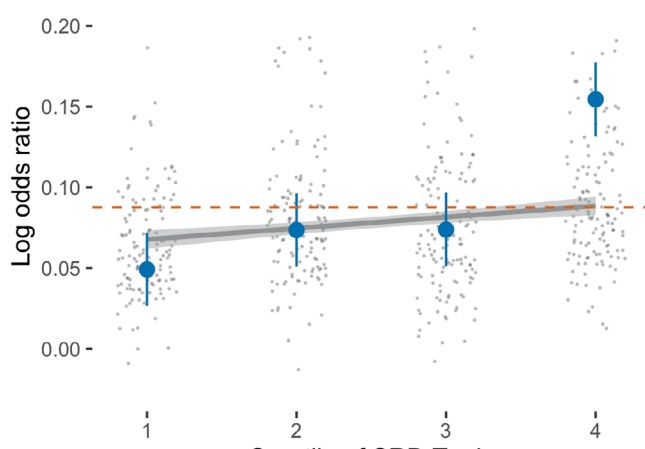

C · rs3735533 x CVD for Essential hypertension
chr7: 7p15.2 (nearest gene: *HOTTIP*)
P = 0.0015 for interaction
P = 0.1 for top/bottom difference
N = 281,367

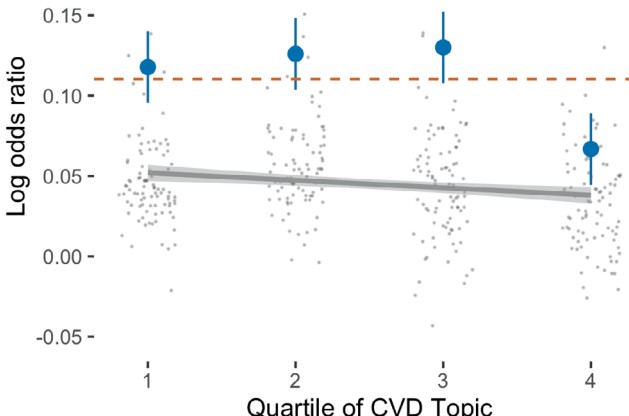

D · rs9404989 x FGND for Hypothyroidism NOS
chr6: 31469789 (nearest gene: *HCG26*)
P = 1e-4 for interaction
P = 3e-03 for top/bottom difference
N = 50,517

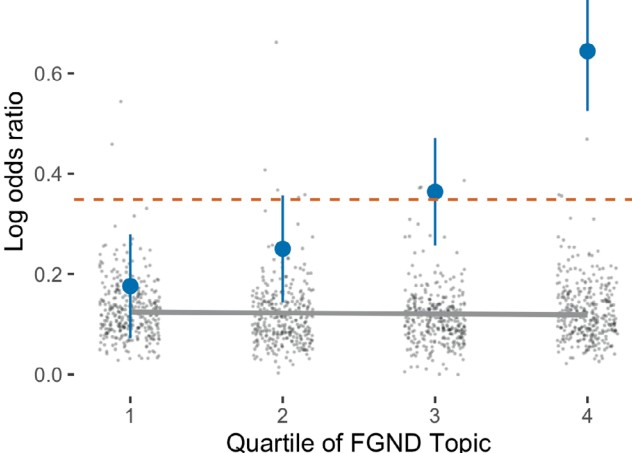

**Extended Data Fig. 8 | Examples of SNP × topic interaction effects on disease phenotypes.** For each example, we report main SNP effects (log odds ratios) specific to each quartile of topic weights across individuals, for both the focal SNP (blue dots) and background SNPs for that disease and topic (genome-wide significant main effect ($P < 5 \times 10^{-8}$) but non-significant SNP × topic interaction effect ($P > 0.05$); gray dots). Dashed red lines denote aggregate main SNP effects for each focal SNP. Error bars denote 95% confidence intervals.

Gray lines denote linear regression of gray dots, with gray shading denoting corresponding 95% confidence intervals. *P*-values for interaction are for testing the interaction regression coefficients; *P*-values for top/bottom differences are for two-sided *t*-test. Panels A–D are examples for type 2 diabetes, asthma, hypertension and hypothyroidism, respectively. Numerical results are reported in Supplementary Table 18.

# Reporting Summary

## Statistics

For all statistical analyses, confirm that the following items are present in the figure legend, table legend, main text, or Methods section.

| n/a | Confirmed | |
|---|---|---|
| ☐ | ☒ | The exact sample size (*n*) for each experimental group/condition, given as a discrete number and unit of measurement |
| ☐ | ☒ | A statement on whether measurements were taken from distinct samples or whether the same sample was measured repeatedly |
| ☐ | ☒ | The statistical test(s) used AND whether they are one- or two-sided *Only common tests should be described solely by name; describe more complex techniques in the Methods section.* |
| ☐ | ☒ | A description of all covariates tested |
| ☐ | ☒ | A description of any assumptions or corrections, such as tests of normality and adjustment for multiple comparisons |
| ☐ | ☒ | A full description of the statistical parameters including central tendency (e.g. means) or other basic estimates (e.g. regression coefficient) AND variation (e.g. standard deviation) or associated estimates of uncertainty (e.g. confidence intervals) |
| ☐ | ☒ | For null hypothesis testing, the test statistic (e.g. *F*, *t*, *r*) with confidence intervals, effect sizes, degrees of freedom and *P* value noted *Give P values as exact values whenever suitable.* |
| ☐ | ☒ | For Bayesian analysis, information on the choice of priors and Markov chain Monte Carlo settings |
| ☐ | ☒ | For hierarchical and complex designs, identification of the appropriate level for tests and full reporting of outcomes |
| ☐ | ☒ | Estimates of effect sizes (e.g. Cohen's *d*, Pearson's *r*), indicating how they were calculated |

*Our web collection on statistics for biologists contains articles on many of the points above.*

## Software and code

Policy information about availability of computer code

| Data collection | N/A |
|---|---|
| Data analysis | Open-source software implementing the ATM method is available at https://github.com/Xilin-Jiang/ATM. BOLT-LMM 2.3 is available at https://alkesgroup.broadinstitute.org/BOLT-LMM/. Heritability and genetic correlation analysis were performed using LDSC, which is available at https://github.com/bulik/ldsc. PLINK v1.9, which was used for FST and association tests, is available at https://www.cog-genomics.org/plink/. |

For manuscripts utilizing custom algorithms or software that are central to the research but not yet described in published literature, software must be made available to editors and reviewers. We strongly encourage code deposition in a community repository (e.g. GitHub). See the Nature Portfolio guidelines for submitting code & software for further information.

## Data

All manuscripts must include a <u>data availability statement</u>. This statement should provide the following information, where applicable:

- Accession codes, unique identifiers, or web links for publicly available datasets
- A description of any restrictions on data availability
- For clinical datasets or third party data, please ensure that the statement adheres to our <u>policy</u>

UK Biobank data is publicly available at https://www.ukbiobank.ac.uk/; All of Us data is publicly available at https://allofus.nih.gov; LD-scores and HAPMAP3 SNP list are available at https://data.broadinstitute.org/alkesgroup/LDSCORE.

## Human research participants

Policy information about <u>studies involving human research participants and Sex and Gender in Research.</u>

| | |
|---|---|
| Reporting on sex and gender | All individuals in UK Biobank are included in the analyses, where we used biological sex that are available in UK Biobank. |
| Population characteristics | N/A |
| Recruitment | N/A |
| Ethics oversight | All individuals in UK Biobank are included in the main analyses; for several secondary analyses we constraint our analysis on British Isle ancestry label provided in UK Biobank, which is stated in the text. |

Note that full information on the approval of the study protocol must also be provided in the manuscript.

# Field-specific reporting

Please select the one below that is the best fit for your research. If you are not sure, read the appropriate sections before making your selection.

☒ Life sciences        ☐ Behavioural & social sciences        ☐ Ecological, evolutionary & environmental sciences

For a reference copy of the document with all sections, see nature.com/documents/nr-reporting-summary-flat.pdf

# Life sciences study design

All studies must disclose on these points even when the disclosure is negative.

| | |
|---|---|
| Sample size | 500,000 UK Biobank participant. No sample size calculation was performed. |
| Data exclusions | No data were excluded from the main analyses; for several secondary analyses we constraint our analysis on British Isle ancestry label provided in UK Biobank, which is stated in the text. |
| Replication | All attempts at replication were successful. |
| Randomization | We used All of Us data to verify the disease topics identified in UK Biobank. We used publicly available data. No participant was recruited. |
| Blinding | No participant was recruited; the investigators were blinded to group allocation during data collection and/or analysis. |

# Reporting for specific materials, systems and methods

We require information from authors about some types of materials, experimental systems and methods used in many studies. Here, indicate whether each material, system or method listed is relevant to your study. If you are not sure if a list item applies to your research, read the appropriate section before selecting a response.

## Materials & experimental systems

| n/a | Involved in the study |
|-----|----------------------|
| ☒ | ☐ Antibodies |
| ☒ | ☐ Eukaryotic cell lines |
| ☒ | ☐ Palaeontology and archaeology |
| ☒ | ☐ Animals and other organisms |
| ☒ | ☐ Clinical data |
| ☒ | ☐ Dual use research of concern |

## Methods

| n/a | Involved in the study |
|-----|----------------------|
| ☒ | ☐ ChIP-seq |
| ☒ | ☐ Flow cytometry |
| ☒ | ☐ MRI-based neuroimaging |

