## [Peer Review File · Nature Genetics]

Peer Review Information

Manuscript Title: Age-dependent topic modelling of comorbidities in UK Biobank identifies disease subtypes with differential genetic risk

Corresponding author name(s): Dr Xilin Jiang, Alkes L. Price, Professor Gil McVean

Reviewer Comments & Decisions:

Decision Letter, initial version:
--

12th January 2023

Dear Dr. Jiang,

Your Article "Age-dependent topic modelling of comorbidities in UK Biobank identifies disease subtypes with differential genetic risk" has been seen by two referees. You will see from their comments below that, while they find your work of potential interest, they have raised substantial concerns that must be addressed. In light of these comments, we cannot accept the manuscript for publication at this time, but we would be interested in considering a suitably revised version that addresses the referees' concerns.

We hope you will find the referees' comments useful as you decide how to proceed. If you wish to submit a substantially revised manuscript, please bear in mind that we will be reluctant to approach the referees again in the absence of major revisions.

To guide the scope of the revisions, the editors discuss the referee reports in detail within the team, including with the chief editor, with a view to identifying key priorities that should be addressed in revision, and sometimes overruling referee requests that are deemed beyond the scope of the current study. In this case, we particularly ask that you address all technical queries related to the methodology and extend the analyses to additional cohorts with detailed electronic health records to further demonstrate the performance and utility of the approach. We hope you will find this prioritized set of referee points to be useful when revising your study. Please do not hesitate to get in touch if you would like to discuss these issues further.

If you choose to revise your manuscript taking into account all reviewer and editor comments, please highlight all changes in the manuscript text file. At this stage we will need you to upload a copy of the manuscript in MS Word .docx or similar editable format.

We are committed to providing a fair and constructive peer-review process. Do not hesitate to contact

us if there are specific requests from the reviewers that you believe are technically impossible or unlikely to yield a meaningful outcome.

*2) If you have not done so already, please begin to revise your manuscript so that it conforms to our Article format instructions, available [here](http://www.nature.com/ng/authors/article_types/index.html). Refer also to any guidelines provided in this letter.

[redacted]

If you wish to submit a suitably revised manuscript, we would hope to receive it within 3-6 months. If you cannot send it within this time, please let us know. We will be happy to consider your revision so long as nothing similar has been accepted for publication at Nature Genetics or published elsewhere. Should your manuscript be substantially delayed without notifying us in advance and your article is eventually published, the received date would be that of the revised, not the original, version.

Nature Genetics is committed to improving transparency in authorship. As part of our efforts in this direction, we are now requesting that all authors identified as 'corresponding author' on published papers create and link their Open Researcher and Contributor Identifier (ORCID) with their account on the Manuscript Tracking System (MTS), prior to acceptance. ORCID helps the scientific community achieve unambiguous attribution of all scholarly contributions. You can create and link your ORCID from the home page of the MTS by clicking on 'Modify my Springer Nature account'. For more information, please visit www.springernature.com/orcid.

Thank you for the opportunity to review your work.

Sincerely,
Kyle

Kyle Vogan, PhD
Senior Editor
Nature Genetics
<https://orcid.org/0000-0001-9565-9665>

Referee expertise:

Referee #1: Genetics, complex traits, statistical methods

Referee #2: Genetics, complex traits, bioinformatics

Reviewers' Comments:

Reviewer #1:
Remarks to the Author:

This paper presents an interesting approach to identify disease subtypes based on biobank-scale data that should be of interest to many readers. I have the following comments/suggestions/questions for the authors' consideration.

1. Data reduction approaches have the potential to increase power to identify disease subtypes. The potential downside with data reduction is one of interpretability. Once we have a significant disease topic, what does it really mean, and how does ambiguity in this regard impact utility?

2. Data issues: Why 282,957 UKB participants instead of the whole UKB or some larger subset? We should not have to wait until deep into the methods to know that it was because you "analysed comorbidity data from 282,957 UK Biobank samples with diagnoses for at least two of the 348 focal diseases that we studied." The fact you have decided to include all 348 diseases with at least 1000 incident cases could be made more clear in the abstract by adding "all" to the relevant sentence.

3. 203-204: How do you define optimal in the statement "The optimal ATM model structure included 10 topics and modelled age-dependent topic loadings for each disease as a spline function with one knot" (see below).

4. 352-355: How should we interpret the significance of "T2D-associated SNP rs1063192 in the CDKN2B locus has a higher odds ratio in the top quartile of cardiovascular topic weight (1.19 ± 0.02) than in the bottom quartile (1.08 ± 0.02) ($P = 4 \times 10^{-5}$ for difference)" given the number of tests? Is that number 348? It should be made clear.

5. Choices are made in the Methods without clear justification or statement that the results are robust to these (arbitrary) choices:

a. 505-506: "The functionals (t) considered are linear, quadratic, cubic polynomials, and cubic splines with one, two and three knots." How do you decide which to use? How likely is it that this set of choices substantially impacted your analyses?

b. 552: we used $\alpha = 1$

c. Should we be concerned that type assignments based on 50.1% and 99.9% are "the same" and the lack of the assignment at 49.9% and assignment at 50.1% are different?

d. Simulations of ATM method: why are these simulation assumptions reasonable and sufficient to explore the range of possible models?

I expect these choices are reasonable and results are robust to the choices, but it would be good to address this directly in the text.

6. The Methods section is not as well written as the remainder of the paper. It is understandable that there are missing details since there is a lot going on. However, the writing itself should be improved. Some examples (many minor):

a. 491: What is the softmax function? I think you define it in the next line, but the connection is not clear.

b. 514: "The details of the inference is explained in Supplementary Note"; is should be are

c. 514-518: the "could"s and "may be considered" make it unclear what you have chosen to do

d. 527-528: "The most commonly used form of $q(z, \theta)$ assume the distribution is factorized"; "assume" should be "assumes"

e. 566: AURPC should be defined at first use

f. 593: "Mathematical expression of ELBO for ATM is presented in equation 9 in Supplementary Note." Are you missing some definite articles?

g. 594: Not sure what this means: "topic numbers between 5 to 20, 25, 30, and 50 topics"

h. 713-721: Map or mapped? Any reason not to be consistent?

i. 719-724: I am not sure what to make of "When a single ICD-10/ICD-10CM code s mapped to more than one PheCodes, we only kept the Phecode that are mapped to the most ICD-10 codes (i.e. PheCode is constructed by combining ICD-10 that represent similar diseases. The Phecode that represent a larger number of ICD-10 codes are more likely to be a well defined disease, which we chose to keep.), which ensure that one ICD-10(CM) code only maps to one PheCode."

j. 740-743: I am not sure what to make of “Most of comorbidity topics are named using the first three topics (e.g. CER: cardiovascular, endocrine/metabolic, respiratory), except for topics that are predominantly associated with one system (LGI: lower gastrointestinal; UGI: upper gastrointestinal; CVD: cardiovascular).”

k. 766-768: I am confused by the numbers listed in this sentence, since they do not correspond to anything mentioned previously: “For all analyses except BOLT-LMM we use 488,377 UK Biobank participants. For BOLT-LMM analyses, we constrain our analysis to 409,694 British Isle ancestry individuals to remove the possibility that topics are capturing population structure.” And what are “BOLT-LMM analyses”? Please give the method as well as the software.

l. 755-756: Restate “to exclude the possibility of population stratification, we compute PRS using mixed-effect association on the British Isle ancestry group.” You cannot exclude the possibility, you can (and do) address it via your analytic strategy.

m. 778-779: “For the computation of PRS, we randomly sampled half of the British isle ancestry population (N = 204,847) for computation efficiency (essential hypertension, arthropathy, asthma, and hypercholesterolemia)”; again, I am unsure what you are saying.

n. 814-815: “We focus on 71 disease and 18 disease subtypes that have heritability z-score above 4 for genetic correlation analysis.” Are the 71 a proper subset of the 348?

o. 855-856: “We used QQ plots to check that the test statistics are well calibrated for each disease-topic pair.” Is it logical to assume that interaction test statistics are well calibrated to the null given the very real possibility that interaction is actually quite common?

p. 884: Why did you simulate variants with MAF randomly sampled from $Uni(0, 0.5)$ when you have data that would allow you to do something more realistic?

Reviewer #2:

Remarks to the Author:

The authors start the manuscript with a statement about “electronic health records (EHR)” but apply their model to a small dataset (in the EHR universe). The authors analysed 282,957 unique UK individuals. The UK BioBank is a very useful genetic resource, but it is not a proper electronic health records database. This is because it lacks temporal health information and involves patient self-reported diagnoses. The authors mention the data includes 1,726,144 diagnoses, so the mean number of diagnoses per individual is 6.1, which is rather small.

The age component of the model introduced by authors is a potential novelty. However, it is unclear if this innovation improves subtyping of diabetes. A plain vanilla topic modeling would surely separate diseases of 40-year-old participants from 69-year-old ones, just because older-age conditions, such as osteoporosis, would be exceedingly rare in the younger participants. Possibly, the authors could think of a quantitative way of convincing the reader that the age component in their model makes a difference.

The authors produce results using a single cohort, making no attempt to replicate or validate their findings. Declaring significant difference of polygenic risk scores across 18 discovered subtypes is a bit puzzling way to quantify significance. There are numerous partitions of the cohort that would produce (statistically) significantly different scores, but this difference is not necessarily biomedically meaningful.

Author Rebuttal to Initial comments

Reviewers' Comments:

Reviewer #1:

Remarks to the Author:

This paper presents an interesting approach to identify disease subtypes based on biobank-scale data that should be of interest to many readers. I have the following comments/suggestions/questions for the authors' consideration.

We thank the reviewer for their positive comments.

1. Data reduction approaches have the potential to increase power to identify disease subtypes. The potential downside with data reduction is one of interpretability. Once we have a significant disease topic, what does it really mean, and how does ambiguity in this regard impact utility?

We agree that interpretability can be a potential downside of data reduction approaches. The interpretation of a particular disease topic is that it consists of diseases that tend to co-occur with a specified set of diseases as a function of age. Identifying the functional biology underlying these co-occurrences remains a direction for future research, but there is immediate utility in performing disease subtype-specific GWAS and downstream analyses using the subtypes that we have identified. We have updated the Discussion section (page 14) to clarify these points.

2. Data issues: Why 282,957 UKB participants instead of the whole UKB or some larger subset? We should not have to wait until deep into the methods to know that it was because you "analysed comorbidity data from 282,957 UK Biobank samples with diagnoses for at least two of the 348 focal

diseases that we studied.” The fact you have decided to include all 348 diseases with at least 1000 incident cases could be made more clear in the abstract by adding “all” to the relevant sentence.

We have updated the *Overview of Methods* subsection of the Results section (page 4) to state that the targeted 282,957 individuals are those with at least two of the 348 diseases studied. We have modified the Abstract (page 1) to clarify that we are studying all 348 diseases that have more than 1,000 occurrences in the HES data.

3. 203-204: How do you define optimal in the statement “The optimal ATM model structure included 10 topics and modelled age-dependent topic loadings for each disease as a spline function with one knot” (see below).

The optimal ATM model refers to the ATM model with the number of topics producing the highest prediction odds ratio on the testing data. We have updated the *Age-dependent disease topic loadings capture comorbidity profiles in the UK Biobank* subsection of the Results section (page 6) to clarify this.

We have updated the ATM software package (<https://github.com/Xilin-Jiang/ATM>) to allow users to compute prediction odds ratios to select optimal ATM model structures. Detailed documentation is available in the “Inferring disease topics using diagnosis data” section of the online README.md file.

4. 352-355: How should we interpret the significance of “T2D-associated SNP rs1063192 in the CDKN2B locus has a higher odds ratio in the top quartile of cardiovascular topic weight (1.19 ± 0.02) than in the bottom quartile (1.08 ± 0.02) ($P = 4 \times 10^{-5}$ for difference)” given the number of tests? Is that number 348? It should be made clear.

We performed both interaction tests and top/bottom quartile tests across 2,530 SNP x Topic pairs spanning 888 disease-associated SNPs, 14 diseases, and 35 disease subtypes. We have updated the Abstract to discuss a different example: the T2D-associated SNP rs1042725 in the HMG2A2 locus has a higher odds ratio in the top quartile of cardiovascular topic weight (1.18 ± 0.02) than in the bottom quartile (1.00 ± 0.02) ($P = 3 \times 10^{-4}$ for interaction test, $P = 3 \times 10^{-7}$ for top/bottom quartile test). In detail, in this example, the interaction test produced a P-value of 3×10^{-4} (which is not Bonferroni significant ($P > 0.05/2,530$) but is FDR-significant ($FDR = 0.04 < 0.1$)) and the test for different odds ratios in top vs.

bottom quartiles (which is a more intuitive test but often less powerful) produced a P-value of 3×10^{-7} (which is Bonferroni significant ($P < 0.05/2,530$) and FDR-significant ($FDR = 0.0002 < 0.1$)). For simplicity, we have elected to report only the $P=3 \times 10^{-7}$ for the top/bottom quartile test ($FDR = 0.0002 < 0.1$) in the Abstract. We have updated the *Disease-associated SNPs have subtype-dependent effects* subsection of the Results section (page 11-12) to report P-values and FDR for both the interaction test and the top/bottom quartile test. We have updated Figure 8A (formerly Figure 7A) accordingly, and have updated all panels of Figure 8 (formerly Figure 7) to report P-values for both the interaction and top/bottom quartile tests.

We note that the analogous results for the example previously discussed in the Abstract are as follows: the T2D-associated SNP rs1063192 in the CDKN2B locus has a higher odds ratio in the top quartile of cardiovascular topic weight (1.18 ± 0.02) than in the bottom quartile (1.19 ± 0.02) ($P=4 \times 10^{-7}$ for interaction test, $P=4 \times 10^{-3}$ for top/bottom quartile test). In detail, in this example, the interaction test produced a P-value of 4×10^{-7} (which is Bonferroni significant ($P < 0.05/2,530$) and FDR-significant ($FDR = 0.0004 < 0.1$)) and the test for different odds ratios in top vs. bottom quartiles (which is a more intuitive test but often less powerful) produced a P-value of 4×10^{-3} (which is not Bonferroni significant ($P > 0.05/2,530$) or FDR-significant ($FDR = 0.17 > 0.1$)). We had previously incorrectly reported the top/bottom quartile test P-value for rs1063192 as 4×10^{-5} in the Abstract and 4×10^{-4} in the Results section, instead of the correct value of 4×10^{-3} . As noted above, we have replaced this example with the T2D-associated SNP rs1042725 in the HMGA2 locus.

5. Choices are made in the Methods without clear justification or statement that the results are robust to these (arbitrary) choices:

We thank the reviewer for pointing out the need for clear justifications in the Methods section. We respond to each point in turn.

a. 505-506: "The functionals (t) considered are linear, quadratic, cubic polynomials, and cubic splines with one, two and three knots." How do you decide which to use? How likely is it that this set of choices substantially impacted your analyses?

The choices of functional form and the number of topics were selected using prediction odds ratio. We evaluated different functional forms (linear, quadratic, cubic polynomials, and cubic splines with one, two and three knots) with increasing flexibility. We determined that all nonlinear functionals (3-7 d.f.) performed similarly (outperforming the linear functional; 2 d.f.) (Supplementary Figure 7), suggesting that these functional forms are flexible enough to fit the data and that the precise choice of functional form evaluated does not substantially impact our analyses. We have updated the Methods Section (page 17) to clarify these points.

b. 552: we used $\alpha = 1$

We have clarified (Methods Section, page 19) that the selection of hyperparameter α puts an uninformative prior on the topic weight distribution, and that the Dirichlet distribution with $\alpha = 1$ has uniform density on the parameter support of the Dirichlet distribution.

c. Should we be concerned that type assignments based on 50.1% and 99.9% are “the same” and the lack of the assignment at 49.9% and assignment at 50.1% are different?

The reviewer makes a good point that discretizing continuous data loses information and may compromise power. We have updated the Limitations paragraph of the Discussion Section (page 14) to note this limitation, while also noting that definitions of disease often discretize continuous variables (e.g. see Falconer 1967). We have updated the *Disease subtypes defined by distinct topics are genetically heterogeneous* subsection of the Results section (page 9) and the Methods Section (page 16) to cite this content.

We have also updated the *Disease subtypes defined by distinct topics are genetically heterogeneous* subsection of the Results Section (page 9) to clarify that some of our disease subtype analyses use continuous-valued topic weights while other disease subtype analyses use discrete subtypes. In detail (in the order of appearance in the text):

- (1) Our PRS analysis uses continuous-valued topic weights;
- (2) Our excess genetic correlation analysis uses discrete subtypes;
- (3) Our excess F_{ST} analysis uses discrete subtypes

(4) Our SNP x Topic interaction analysis uses continuous-valued topic weights.

d. Simulations of ATM method: why are these simulation assumptions reasonable and sufficient to explore the range of possible models?

In our simulations of the ATM method we aimed to choose simulation parameters that resemble real data, matching the values of average number of disease diagnoses per individual, ratio of #individuals/#diseases, topic loadings, and standard deviation in age at diagnosis observed in real data. As multiple parameter settings match those values, we performed sensitivity analyses in simulations with different values of population size, average number of diseases per individual, number of distinct diseases, and number of underlying disease topics, confirming that results were not sensitive to these choices (Supplementary Figure 4 and Supplementary Figure 5). We have updated the Simulations Section (page 5) to clarify these points. Thus, all parameters were either specified to resemble real data or varied in sensitivity analyses.

I expect these choices are reasonable and results are robust to the choices, but it would be good to address this directly in the text.

We thank the reviewer for suggesting that the choices are reasonable and the results are robust to these choices. We have addressed each of these choices directly in the text.

6. The Methods section is not as well written as the remainder of the paper. It is understandable that there are missing details since there is a lot going on. However, the writing itself should be improved. Some examples (many minor):

We thank the reviewer for flagging these details. We respond to each point in turn.

a. 491: What is the softmax function? I think you define it in the next line, but the connection is not clear.

We have updated the Methods Section (page 17) to clarify why we use the softmax function.

b. 514: “The details of the inference is explained in Supplementary Note”; is should be are

We have fixed this typo in the Methods section (page 17).

c. 514-518: the “could”s and “may be considered” make it unclear what you have chosen to do

We have updated this text in the Methods Section (page 18) to clarify our chosen inference method, prioritizing the text “is” in preference to “could be”.

d. 527-528: “The most commonly used form of $q(z, \theta)$ assume the distribution is factorized”; “assume” should be “assumes”

We have fixed this typo in the Methods section (page 18).

e. 566: AURPC should be defined at first use

We have added a definition of the acronym AUPRC to the Methods section (page 19).

f. 593: “Mathematical expression of ELBO for ATM is presented in equation 9 in Supplementary Note.”
Are you missing some definite articles?

We have fixed this typo in the Methods section (page 20) by changing “Mathematical expression” to “The mathematical expression” and “Supplementary Note” to “the Supplementary Note”.

g. 594: Not sure what this means: “topic numbers between 5 to 20, 25, 30, and 50 topics”

We have updated the Methods section (page 20) to clarify that we evaluated 19 choices of the number of topics: 5-20, 25, 30, and 50.

h. 713-721: Map or mapped? Any reason not to be consistent?

We have fixed this typo in the Methods section (page 23) by using the term “mapped” (instead of “map”) throughout, except when specifically referring to “map files” (noun).

i. 719-724: I am not sure what to make of “When a single ICD-10/ICD-10CM code s mapped to more than one PheCodes, we only kept the Phecode that are mapped to the most ICD-10 codes (i.e. PheCode is constructed by combining ICD-10 that represent similar diseases. The Phecode that represent a larger number of ICD-10 codes are more likely to be a well defined disease, which we chose to keep.), which ensure that one ICD-10(CM) code only maps to one PheCode.”

We have updated this sentence in the Methods section (page 23) to clarify how we chose the Phecode when the mapping between Phecode and ICD-10CM is not one-to-one.

j. 740-743: I am not sure what to make of “Most of comorbidity topics are named using the first three topics (e.g. CER: cardiovascular, endocrine/metabolic, respiratory), except for topics that are predominantly associated with one system (LGI: lower gastrointestinal; UGI: upper gastrointestinal; CVD: cardiovascular).”

We have updated this sentence in the Methods section (page 24) to clarify that most topics are named using the three *most common Phecode disease systems* represented in the diseases underlying the topic.

k. 766-768: I am confused by the numbers listed in this sentence, since they do not correspond to anything mentioned previously: “For all analyses except BOLT-LMM we use 488,377 UK Biobank participants. For BOLT-LMM analyses, we constrain our analysis to 409,694 British Isle ancestry

individuals to remove the possibility that topics are capturing population structure.” And what are “BOLT-LMM analyses”? Please give the method as well as the software.

We have updated this part of the Methods section (page 26) to clarify that PRS analyses using the BOLT-LMM mixed model method and software (ref. 50-51; see Code Availability) restrict to 409,695 samples of British Isle ancestry (in the union of training and test data), whereas ATM, genetic correlation analysis, F_{ST} , and SNP x Topic interaction analyses use 488,377 UK Biobank samples (prior to restricting to 282,957 samples with at least two of the 348 diseases studied).

l. 755-756: Restate “to exclude the possibility of population stratification, we compute PRS using mixed-effect association on the British Isle ancestry group.” You cannot exclude the possibility, you can (and do) address it via your analytic strategy.

We have modified this part of the Methods section (page 26) to state that in PRS analyses we applied the BOLT-LMM mixed model method and software (ref. 50-51; see Code Availability) to samples of British Isles ancestry in order to adjust for and minimize the impact of population stratification, while noting that the possibility of population stratification cannot be excluded (Haworth et al. 2019 Nat Commun).

m. 778-779: “For the computation of PRS, we randomly sampled half of the British isle ancestry population (N = 204,847) for computation efficiency (essential hypertension, arthropathy, asthma, and hypercholesterolemia)”;

again, I am unsure what you are saying.

We have rewritten this sentence in the Methods section (Page 27) to clarify that we downsample to half of the controls for computation efficiency, which has little impact on sampling noise since sampling noise is proportional to $1/N_{\text{case}} + 1/N_{\text{control}}$ (where N_{control} is generally much larger than N_{case}).

n. 814-815: “We focus on 71 disease and 18 disease subtypes that have heritability z-score above 4 for genetic correlation analysis.” Are the 71 a proper subset of the 348?

The reviewer is correct that the 71 diseases are a subset of the 348 diseases studied. We have updated the Methods section (page 28) to clarify this point.

We further note that the 18 disease subtypes are a subset of the 30 disease subtypes studied. We have updated the Methods section (page 28) to clarify this point.

o. 855-856: “We used QQ plots to check that the test statistics are well calibrated for each disease-topic pair.” Is it logical to assume that interaction test statistics are well calibrated to the null given the very real possibility that interaction is actually quite common?

We have updated the Methods section (page 29) to clarify that we used QQ plots to check that interaction test statistics computed using all non-subtype topics for each disease (which are expected to be null; see below) were well-calibrated. Results are reported in Supplementary Figure 31 (formerly Supplementary Figure 25), cited in the *Disease-associated SNPs have subtype-dependent effects* subsection of the Results section (page 12; this text previously erroneously cited Supplementary Figure 24, we regret the error).

In detail, the test statistics that we computed for calibration purposes are for SNP x Topic interaction between GWAS-SNP and non-subtype topics, which are expected to be well-calibrated under the null even if interaction involving disease subtypes is common. We have also clarified in the Supplementary Figure 31 that the median p-values for these null tests was 0.47, compared to 0.35 for the disease-subtype topic interaction tests. The small inflation in test statistics ($0.47 < 0.5$) may be caused by the correlation between topics (i.e. a SNP that interacts with a subtype-topic is expected to have weak interaction with other non-subtype topics as the topic weights sum to one). We have clarified this point in the caption of Supplementary Figure 31 (formerly Supplementary Figure 25).

p. 884: Why did you simulate variants with MAF randomly sampled from $Uni(0, 0.5)$ when you have data that would allow you to do something more realistic?

The reviewer makes a good point that it is possible to use the empirical MAF distribution in this experiment. We have now performed this analysis. Results using the empirical MAF distribution were

virtually unchanged, and are now reported in Supplementary Figure 28 (Formerly Supplementary Figure 22). We have updated the Methods section accordingly (page 30).

Reviewer #2: (numbers added to reviewer comments)

1. The authors start the manuscript with a statement about “electronic health records (EHR)” but apply their model to a small dataset (in the EHR universe). The authors analysed 282,957 unique UK individuals. The UK BioBank is a very useful genetic resource, but it is not a proper electronic health records database. This is because it lacks temporal health information and involves patient self-reported diagnoses. The authors mention the data includes 1,726,144 diagnoses, so the mean number of diagnoses per individual is 6.1, which is rather small.

The reviewer has raised two related concerns: (i) UK Biobank is a small data set (in the EHR universe), with only 282,957 individuals and 6.1 diagnoses per individual; and (ii) UK Biobank lacks temporal health information and involves patient self-reported diagnoses. We address each of the concerns in turn.

(i) UK Biobank is a small data set (in the EHR universe), with only 282,957 individuals and 6.1 diagnoses per individual.

We have addressed this concern in two ways.

First, we have now applied ATM to a new data set, All of Us, with 211,908 U.S. individuals and 14.6 diagnoses per individual. All of Us has almost twice as many diagnoses as the UK Biobank data set, almost tripling the number of diagnoses analysed in this study. The concordance between UK Biobank and All of Us results provides an important validation (see response to Reviewer #2 Comment 3 for details). We have added a new *Age-dependent disease topic loadings capture concordant comorbidity profiles in All of Us* subsection of the Results section (page 8-9, new Figure 5 and new Supplementary Figure 15-19, 22, new Supplementary Table 6-9) to report these results, and have updated the Abstract (page 2) and Discussion section (page 13-14) to discuss the new results.

Second, we have updated the Discussion section (page 13) to note the potential for applying ATM to much larger data sets such as CVD-COVID-UK (Ross 2021 BMJ), after considering its computational cost; ATM required 4.6 hours for the UK Biobank analyses, and its running time scales linearly with the total number of diagnoses (Supplementary Table 20, formerly Supplementary Table 16).

(ii) UK Biobank lacks temporal health information and involves patient self-reported diagnoses.

Our initial description of the resource was insufficiently clear. We have updated the *Age-dependent disease topic loadings capture comorbidity profiles in the UK Biobank* subsection of the Results section (page 6) and Methods section (page 23) to provide two clarifications. First, UK Biobank includes temporal health information, with hospital records spanning an average of 28.6 years in the individuals that we analysed. Second, we exclusively analyse hospital records that do not include patient self-reported diagnoses. We have included analogous statements about All of Us in the Methods section (page 24-25).

2. The age component of the model introduced by authors is a potential novelty. However, it is unclear if this innovation improves subtyping of diabetes. A plain vanilla topic modeling would surely separate diseases of 40-year-old participants from 69-year-old ones, just because older-age conditions, such as osteoporosis, would be exceedingly rare in the younger participants. Possibly, the authors could think of a quantitative way of convincing the reader that the age component in their model makes a difference.

We agree that the age component of ATM is an innovation, and we agree that it is important to assess the impact of this innovation on our results in real data (in addition to our extensive simulation results comparing ATM to LDA [which does not model age] in Figure 2 and Supplementary Figures 2-4). Previously, we stated that, in our UK Biobank analyses, “ATM attained higher prediction odds ratios than LDA [which does not model age] across different values of the number of topics (Supplementary Figure 8)”. We have now updated this text (*Age-dependent disease topic loadings capture comorbidity profiles in the UK Biobank* subsection of the Results section, page 7) to report that ATM attained an average prediction odds ratio of 1.71, compared to a prediction odds ratio of 1.58 for LDA. We have elected to keep Supplementary Figure 8 as a supplementary figure given the large number of analyses in this manuscript, but are open to making it a main Figure panel if reviewers and/or editors express a strong preference.

3. The authors produce results using a single cohort, making no attempt to replicate or validate their findings. Declaring significant difference of polygenic risk scores across 18 discovered subtypes is a bit puzzling way to quantify significance. There are numerous partitions of the cohort that would produce (statistically) significantly different scores, but this difference is not necessarily biomedically meaningful.

The reviewer has raised two related concerns: (i) we produce results using a single cohort, without validating our findings; and (ii) a significant difference in polygenic risk scores across subtypes is not necessarily biomedically meaningful, as there many partitions of the cohort that would produce this result. We address each of these concerns in turn.

(i) we produce results using a single cohort, without validating our findings.

We agree that there is immense value in analysing data from a new cohort to validate our findings. Thus, we have now applied ATM to a new data set, All of Us, with 211,908 U.S. individuals and 14.6 diagnoses per individual. We have added a new *Age-dependent disease topic loadings capture concordant comorbidity profiles in All of Us* subsection of the Results section (page 8-9, new Figure 5 and new Supplementary Figure 15-19, 22, new Supplementary Table 6-9) to report these results, and have updated the Discussion section (page 13-14) to discuss the new results. The concordance between UK Biobank and All of Us results provides an important validation of our findings. We summarize 3 of our most important conclusions:

First, the 10 UK Biobank topics align well with the 13 All of Us topics (#topics in each cohort chosen to maximize prediction odds ratio). For each UK Biobank topic, we computed the Spearman correlation of topic loadings for each disease (averaged across ages) with each All of Us topic; the median Spearman correlation with the most similar All of Us topic was equal to 0.54. See page 8, Figure 5B, Supplementary Figure 18, and two examples (CER and CVD topics) in Figure 5A.

Second, topic loadings inferred from All of Us enable us to predict disease in UK Biobank using comorbidity information. The prediction odds ratio in UK Biobank was equal to 1.32 (s.e. = 0.0027) (as compared to 1.71 using UK Biobank topics in held-out UK Biobank samples). See *Age-dependent disease topic loadings capture concordant comorbidity profiles in All of Us* subsection of the Results section page 8, citing Supplementary Figure 16C. We note 3 key differences between All of Us and UK Biobank data: (i) All of Us contains primary care and hospital data encoded using SNOMED clinical terms, whereas UK Biobank uses hospitalization episode statistics (HES; encoded using ICD-10 clinical terms); (ii) All of Us is based on the U.S. population and U.S. health care system whereas UK Biobank is based on the UK population and UK health care system, which impacts diagnostic criteria and age at diagnosis; and (iii) All of Us individuals have different ancestries and socioeconomic backgrounds (including 26% African and 17% Latino; 78% of All of Us is historically underrepresented in biomedical research based on race, ethnicity, age, gender identity, disability status, medical care access, income, and educational

attainment) than UK Biobank individuals (94% European with higher than income and average educational attainment). We consider the cross-cohort prediction odds ratio of 1.32 to be an encouraging result given these key differences.

Third, disease subtypes identified in All of Us align well with subtypes identified in UK Biobank. Comorbidity-derived subtype correlation between UK Biobank and All of Us was 0.70 for all 233 diseases shared between UK Biobank and All of Us and 0.64 for 41 of these 233 diseases that have subtypes. See page 8-9, Table 2, Figure 5C, and Supplementary Figure 19.

(ii) a significant difference in polygenic risk scores across subtypes is not necessarily biomedically meaningful, as there many partitions of the cohort that would produce this result

We are unsure exactly what type of scenario the reviewer is referring to, but we believe that a scenario of particular interest is the scenario in which genetic differences between subtypes (quantified by genetic correlation or F_{ST}) could be unrelated to disease (e.g. we expect a nonzero genetic correlation and nonzero F_{ST} between tall vs. short type 2 diabetes cases, even if height is not genetically correlated to type 2 diabetes). To investigate whether this type of scenario could explain our findings, we performed two analyses. First, we verified that excess genetic correlations between disease-subtype and subtype-subtype pairs (Figure 7 [formerly Figure 6]) were almost unchanged when repeating the genetic correlation analysis using disease cases and controls with matched topic weights (i.e. cases and controls have matched topic weight distributions within each disease or disease subtype; this procedure controlled factors other than the disease under consideration, e.g., height when analysing the T2D subtypes) (Supplementary Figure 26; formerly Supplementary Figure 20). This analysis was previously reported in the *Disease subtypes defined by distinct topics are genetically heterogeneous* subsection of the Results section (page 10), but we have updated the text in an effort to improve clarity. Second, we assessed the statistical significance of F_{ST} estimates between two subtypes of disease cases by comparing these values to F_{ST} estimates between healthy controls with matched topic weights (i.e. F_{ST} estimates between two sets of healthy controls with topic weight distributions matched to the respective disease subtypes, which captured the empirical null F_{ST} discrepancy when disease signal was absent) (Supplementary Figure 27 (formerly Supplementary Figure 21)). This analysis was previously reported in the *Disease subtypes defined by distinct topics are genetically heterogeneous* subsection of the Results section (page 10-11), but we have updated the text in an effort to improve clarity.

Decision Letter, first revision:

18th May 2023

Dear Xilin,

Your revised manuscript "Age-dependent topic modelling of comorbidities in UK Biobank identifies disease subtypes with differential genetic risk" (NG-A61196R1) has been seen by the original referees. As you will see from their comments below, they find that the paper has improved in revision, and therefore we will be happy in principle to publish it in Nature Genetics as an Article pending final revisions to comply with our editorial and formatting guidelines.

We are now performing detailed checks on your paper, and we will send you a checklist detailing our editorial and formatting requirements soon. Please do not upload the final materials or make any revisions until you receive this additional information from us.

Thank you again for your interest in Nature Genetics. Please do not hesitate to contact me if you have any questions.

Sincerely,
Kyle

Kyle Vogan, PhD
Senior Editor
Nature Genetics
<https://orcid.org/0000-0001-9565-9665>

Reviewer #1 (Remarks to the Author):

The authors have addressed my comments well. I have no further issues to raise.

Reviewer #2 (Remarks to the Author):

This reviewer is happy with the revised version of the manuscript.

Final Decision Letter:

31st August 2023

Dear Xilin,

I am delighted to say that your manuscript "Age-dependent topic modelling of comorbidities in UK

Biobank identifies disease subtypes with differential genetic risk" has been accepted for publication in an upcoming issue of Nature Genetics.

Your paper will be published online after we receive your corrections and will appear in print in the next available issue. You can find out your date of online publication by contacting the Nature Press Office (press@nature.com) after sending your e-proof corrections. Now is the time to inform your Public Relations or Press Office about your paper, as they might be interested in promoting its publication. This will allow them time to prepare an accurate and satisfactory press release. Include your manuscript tracking number (NG-A61196R2) and the name of the journal, which they will need when they contact our Press Office.

Before your paper is published online, we will be distributing a press release to news organizations worldwide, which may very well include details of your work. We are happy for your institution or funding agency to prepare its own press release, but it must mention the embargo date and Nature Genetics. Our Press Office may contact you closer to the time of publication, but if you or your Press Office have any enquiries in the meantime, please contact press@nature.com.

Please note that Nature Genetics is a Transformative Journal (TJ). Authors may publish their research with us through the traditional subscription access route or make their paper immediately open access through payment of an article-processing charge (APC). Authors will not be required to make a final decision about access to their article until it has been accepted. <[a href="https://www.springernature.com/gp/open-research/transformative-journals"](https://www.springernature.com/gp/open-research/transformative-journals)> Find out more about Transformative Journals

Authors may need to take specific actions to achieve <[a href="https://www.springernature.com/gp/open-research/funding/policy-compliance-faqs"](https://www.springernature.com/gp/open-research/funding/policy-compliance-faqs)> compliance with funder and institutional open access mandates. If your research

is supported by a funder that requires immediate open access (e.g. according to [Plan S principles](https://www.springernature.com/gp/open-research/plan-s-compliance)) then you should select the gold OA route, and we will direct you to the compliant route where possible. For authors selecting the subscription publication route, the journal's standard licensing terms will need to be accepted, including [self-archiving-and-license-to-publish](https://www.nature.com/nature-portfolio/editorial-policies/self-archiving-and-license-to-publish). Those licensing terms will supersede any other terms that the author or any third party may assert apply to any version of the manuscript.

If you have not already done so, we invite you to upload the step-by-step protocols used in this manuscript to the Protocols Exchange, part of our on-line web resource, natureprotocols.com. If you complete the upload by the time you receive your manuscript proofs, we can insert links in your article that lead directly to the protocol details. Your protocol will be made freely available upon publication of your paper. By participating in natureprotocols.com, you are enabling researchers to more readily reproduce or adapt the methodology you use. [Natureprotocols.com](http://natureprotocols.com) is fully searchable, providing your protocols and paper with increased utility and visibility. Please submit your protocol to <https://protocolexchange.researchsquare.com/>. After entering your [nature.com](http://www.nature.com) username and password you will need to enter your manuscript number (NG-A61196R2). Further information can be found at <https://www.nature.com/nature-portfolio/editorial-policies/reporting-standards#protocols>

Sincerely,
Kyle

Kyle Vogan, PhD

Senior Editor
Nature Genetics
<https://orcid.org/0000-0001-9565-9665>